# Atmospheric teleconnections between the Arctic and the Eastern Baltic Sea regions

Liisi Jakobson[1,2], Erko Jakobson[2]; Piia Post[1], Jaak Jaagus[1]

[1]University of Tartu, Tartu, Estonia

[2]Tartu Observatory, Tõravere, Estonia

*Correspondence to:* Liisi Jakobson (Liisi.Jakobson@ut.ee)

**Abstract.** The teleconnections between meteorological parameters of the Arctic and the Eastern Baltic Sea regions were analysed based on the NCEP-CFSR and ERA-Interim reanalysis data for 1979–2015. The Eastern Baltic Sea region was characterised by meteorological values at a testing point (TP) in Southern Estonia (58°N, 26°E). Temperature at the 1000 hPa level at the TP have a strong negative correlation with the Greenland sector (the region between 55–80°N and 20–80°W) during all seasons except summer. Significant teleconnections are present in temperature profiles from 1000 to 500 hPa. The strongest teleconnections between the same parameter at the Eastern Baltic Sea region and the Arctic are found in winter, but they are clearly affected by the Arctic Oscillation (AO) index. After removal of the AO index variability, correlations in winter were below ±0.5, while in other seasons there remained regions with strong ($|R|>0.5$, $p<0.002$) correlations. Strong correlations ($|R|>0.5$) are also present between different climate variables (sea-level pressure, specific humidity, wind speed) at the TP and different regions of the Arctic. These teleconnections cannot be explained solely with the variability of circulation indices. The positive temperature anomaly of mild winter at the Greenland sector shifts towards east during the next seasons, reaching to Scandinavia/Baltic Sea region in summer. This evolution is present at 60°N and 65°N but is missing at higher latitudes. The most permanent lagged correlations in 1000 hPa temperature reveals that the temperature in summer at the TP is strongly predestined by temperature in the Greenland sector in the previous spring and winter.

## 1 Introduction

Over the past half a century, the Arctic has warmed at about twice the global rate (IPCC, 2014), a phenomenon called the Arctic Amplification (AA). At the same time, a significant decrease in sea ice extent has occurred in all calendar months since 1979 (Simmonds, 2015), which has been declared to have a leading role in recent AA by some scientists (e.g. Screen and Simmonds, 2010; Francis and Vavrus, 2012). On the other hand, Perlwitz et al. (2015) disagree the common assumption that sea ice decline is primarily responsible for the amplified Arctic tropospheric warming. They found that from October to December, the main factors responsible for the Arctic deep tropospheric warming are: 1) the recent decadal fluctuations and 2) long-term changes in sea surface temperatures. These two factors are located outside the Arctic. According to Sato et al. (2014) warm southerly advection is favourable for retreating sea ice over the Barents Sea and warming of air aloft, whereas sea-ice decline would result in warming over the Barents Sea because of anomalous turbulent heat fluxes. Screen et al. (2012) found that sea ice concentration and sea surface temperature explain a large portion of the observed Arctic near-surface warming, whereas remote sea surface temperature changes explain the majority of observed warming aloft. As the energy budget of the Arctic is highly dependent on energy exchange with lower latitudes, then the changes in atmospheric and oceanic circulation play an important role in all kinds of heat conservation changes in the Arctic, most prominently expressed in sea ice volume variations. The observed enhanced warming of the Arctic, referred to as the AA, is expected to be related to further changes that impact mid-latitudes and the rest of the world (Jung et al., 2015; Walsh, 2014). These Arctic influences could be

direct, as the advection of cold and dry air from over the ice-covered areas to the neighbouring territories, but it could also be through teleconnections – the large-scale patterns of high and low pressure systems and circulation anomalies that cover vast geographical areas and reflect the non-periodic oscillations of the climate system.

Teleconnection as a term was first used by Ångström (1935). Wallace and Gutzler (1981) defined teleconnections as significant simultaneous correlations between the time series of meteorological parameters at widely separated points on the Earth; the essence of a teleconnection is that a climatic process may influence the Earth's system elsewhere (Liu and Alexander, 2007). Teleconnections between the Arctic and mid-latitude regions have been the focus of research for many years and several reviews about the Arctic sea ice impact on the global climate (Budikova, 2009; Vihma, 2014) or Eurasian climate (Gao et al., 2015) have been published. Budikova (2009) stated that the size of the response of the climate of remote regions to changes in the Arctic has been found to be linked linearly, but the forcing direction nonlinearly. Less ice in the Arctic results in a significant decrease in the speed of the westerlies and the intensities of storms poleward of 45°N (Budikova, 2009). Many studies suggest that the Arctic sea ice decline increases the probability for circulation patterns resembling the negative phase of the Arctic Oscillation (AO) and North Atlantic Oscillation (NAO) indices in winter (Vihma, 2014). However, there is a debate as to whether the reduction in the autumn Arctic sea-ice-induced negative AO/NAO index can persist into winter and suggest that winter atmospheric circulation is more closely associated with changes in the winter Arctic sea ice (Gao et al., 2015). Several studies have demonstrated relationships between warming and/or ice decline, and mid-latitude weather and climate extremes (Handorf et al., 2015; Coumou et al., 2014; Tang et al., 2013; Petoukhov et al., 2013; Francis and Vavrus, 2012; Petoukhov and Semenov, 2010). Others have analysed whether these associations are statistically and/or physically robust (Hassanzadeh et al., 2014; Screen et al., 2014; Barnes et al., 2014; Screen and Simmonds, 2013, 2014; Barnes, 2013), while some investigations suggest that the apparent associations may have their origin, in part, in remote influences (Perlwitz et al., 2015; Sato et al., 2014; Peings and Magnusdottir, 2014; Screen et al., 2012; Petoukhov and Semenov, 2010).

The linkages between the Arctic and midlatitudes depend on geographical region, season and other impacts. There are certain geographical regions in the Arctic that have greater amount of warming and the influence of these is more investigated. Arctic warming over the Barents and Kara Seas and its impacts on the mid-latitude circulations have been widely discussed (Dobricic et al., 2016; Semenov and Latif, 2015; Kug et al., 2015; Sato et al., 2014). Another particular regional warm core (Screen and Simmonds, 2010) is the East Siberian and Chukchi Seas, which is related to severe winters over North America (Kug et al., 2015; Lee et al., 2015). Screen and Simmonds (2010) brought out also the third particular regional warm core – northeast Canada and Greenland which has been less investigated. Wu et al., (2013) focused on winter sea ice concentration west of Greenland, including the Labrador Sea, Davis Strait, Baffin Bay, and Hudson Bay and found that winter sea ice concentration west of Greenland is a possible precursor for summer atmospheric circulation and rainfall anomalies over northern Eurasia. If we look at the regions in the mid-latitudes then potential Arctic teleconnections with Europe are less clear than with North America and Asia (Overland et al., 2015). The linkages between the Arctic and midlatitudes depend also on season. Summer is exceptional season when the weather conditions are less affected by large-scale atmospheric circulation both in midlatitudes and in the Arctic. But the influence of the increase in late summer open water area is directly contributing to a modification of large scale atmospheric circulation patterns (Overland and Wang, 2010).

~~The relationship between the AA and weather extremes and/or persistent weather patterns in mid-latitudes are mostly explained with Arctic and North Atlantic anomalous circulation regimes, waviness and strength of jet stream (Vavrus et al., 2017; Francis and Skific, 2015; Overland et al., 2015; Barnes and Screen, 2015; Francis and Vavrus, 2015; Coumou et al., 2014; Tang et al., 2013; Petoukhov et al., 2013; Francis and Vavrus, 2012). Common supposition is that sea ice declines are primarily responsible for amplified Arctic tropospheric warming. This conjecture is central to a hypothesis in which Arctic sea ice loss forms the beginning link of a causal chain that includes weaker westerlies in mid-latitudes, more persistent and amplified mid-latitude waves, and more extreme weather (Perlwitz et al., 2015). On the other hand Sun et al. (2016) brought out that neither sea ice loss nor anthropogenic forcing overall yields the winter cold extremes and persistence in mid-latitudes. Arctic warming over~~

The Eastern Baltic Sea region features very variable weather conditions due to its location in the climatic transition zone between the North Atlantic and the Eurasian continent. The region is also close to the Arctic or even part of it (depending on defining the borderlines) and certainly has a direct Arctic influence. The weather in the region depends highly on the position of the polar front: it can be located northward as well as southward of the area. According to the Second Assessment of Climate Change for the Baltic Sea Basin (BACC II, 2015), significant changes have occurred in climate parameters, which could be associated to large-scale atmospheric circulation. Intensity of the zonal circulation, i.e. the westerlies, has increased during the cold period (NDJFM), especially in February and March. After 1980s there has been significant temperature increase in the Baltic Sea region (BACC II, 2015), which has not been equal throughout a year. The highest increase in air temperature is typical for spring (MAM) season. Although a remarkable warming has been present also in winter (DJF), it is not significant due to a very high temporal variability (Jaagus, 2006). A tendency of increasing precipitation in winter and spring was detected in the Baltic Sea region during the latter half of the 20th century, possibly increasing the risk of extreme precipitation events. There is some evidence that the intensity of storm surges may have increased in some parts of the Baltic Sea in recent decades, and this has been attributed to long-term shifts in the tracks of some cyclone types rather than to a long-term change in the intensity of storminess (BACC II, 2015).

It is known that fluctuations of the climatological parameters in the Baltic Sea region are strongly affected by the atmospheric circulation variability described by the teleconnection indices of NAO and AO (BACC II, 2015). There is no clear understanding about the reasons for the changes in these indices or climatic parameters in the Baltic Sea region in most recent time. One of the reasons for incomplete understanding is the non-stationarity of the NAO spatial pattern and the temporal correlations (Lehmann et al., 2011; 2017). It is natural to assume that changes in the Arctic climate system have an effect on the Eastern Baltic Sea region and vice versa due to their close proximity. Our aim is to clarify how the climatic parameters in the Eastern Baltic Sea and Arctic regions are associated. Knowledge of such connections helps to define regions in the Arctic that could be with higher extent associated with the Eastern Baltic region climate change.

Our analysis is designed as follows. We selected one grid point (the testing point TP) to represent the Eastern Baltic Sea region and to find correlations between climate parameters at this point and in the Arctic cap to find the regions of mutual connections. Next, we calculate the partial correlations with various teleconnection indices as control factors to get rid of the known atmospheric circulation variability. To compare broad atmospheric circulation patterns, we calculate geopotential heights differences between cold and mild winters. Due to longer memory of non-atmospheric components of climate system we compute by season lagged correlations between climate parameters.

The objective of this paper is to indicate the relationships between meteorological parameters of the Arctic region and the TP. By tracking down the teleconnections between the rapidly changing Arctic region and the TP we can get valuable information about possible future trends in the Eastern Baltic Sea region even if the changes in both regions were caused by a third factor.

## 2 Data and methodology

We used monthly mean reanalysis data from the Climate Forecast System Reanalysis (CFSR) on a 0.5º×0.5º horizontal grids provided by the National Centers for Environmental Prediction (NCEP). For the period 1979–2010, 6-hourly data of CFSR version 1 (Saha et al., 2010) were used for calculating monthly means and for 2011–2015, 6-hourly data of CFSR version 2 were used (Saha et al., 2014). For reproducibility, we repeated all calculations using ERA-Interim data (Dee et al., 2011). Mostly, these two models showed very similar results. In this paper, only NCEP-CFSR results are shown, only disagreements between models are pointed out in the Results chapter. The following parameters were analysed: temperature at 1000, 850, 500 and 250 hPa level, sea-level pressure (SLP), geopotential heights from 1000 hPa to 100 hPa, specific humidity and wind speed at 1000 hPa, and sea ice concentration (SIC). Monthly means and seasonal means (DJF, MAM, JJA, SON) were calculated from the 6-hourly data. Monthly mean wind speed was calculated as a scalar average.

The teleconnection indices we applied in our analyses were chosen according to the possible influence due to the geographical position of the centres of action of the teleconnection patterns over the North-Atlantic-Eurasian region. The following indices were chosen: 1) The North Atlantic Oscillation (NAO), which is the dominant mode of atmospheric variability in the North Atlantic sector throughout the year (Barnston and Livezey, 1987); 2) The Arctic Oscillation (AO), which is usually defined as the first EOF of the mean sea level pressure field in the Northern Hemisphere (Ambaum et al., 2001); 3) The Scandinavian Pattern (SCA), which consists of a primary circulation centre over Scandinavia, with two other weaker centres of action with the opposite sign, one over the north eastern Atlantic and the other over central Siberia to the southwest of Lake Baikal (Bueh and Nakamura, 2007); 4) The East Atlantic Pattern (EA), which consists of a north-south dipole of anomaly centres spanning the North Atlantic from east to west (Barnston and Livezey, 1987); 5) The East Atlantic/West Russia Pattern (EA/WR), which consists of four main anomaly centres: Europe, northern China, central North Atlantic and north of the Caspian Sea; 6) The Polar/ Eurasia Pattern (PEU) consists of height anomalies over the polar region, and opposite anomalies over northern China and Mongolia.; 7) Additionally, Pacific Decadal Oscillation (PDO), which is the dominant year-round pattern of monthly North Pacific sea surface temperature (SST) variability was included. Although its geographical centres are far from the Baltic Sea region, Uotila et al (2015) found that PDO correlated significantly with the ice concentration and temperature of Baltic Sea. All indices were downloaded from the NOAA-CPC database (http://www.cpc.noaa.gov).

We assume that because of a high spatial correlation between meteorological conditions in the Eastern Baltic Sea region it is reasonable to choose one testing point (TP) that represents the region. Lehmann et al. (2011) used the same method by selecting Hamburg-Fuhlsbüttel to represent the temperature evolution of the Baltic Sea area as correlation coefficients for the seasonal mean air temperature between Hamburg and different sub-basins ranged from 0.8 to 0.9. We selected TP in Southern Estonia with coordinates 58°N, 26°E. The correlation coefficient for the seasonal mean air temperature at 1000 hPa and SLP (not shown) between the TP and different sub-basins of the Eastern Baltic Sea is mostly higher than 0.85 (Figure 1). The highest correlation is observed in winter and the lowest in summer.

We analysed the reanalysis data with the Grid Analysis and Display System (GrADS). We calculated linear correlation coefficients to reveal teleconnections between the Arctic region and the TP of the Eastern Baltic Sea region. In this paper we use only linear Pearson correlations, non-linear correlations are not included. We define the Arctic region here as the region northward of 55°N. Larger region than usual (Arctic cap from polar circle or 70°N; July 10°C isotherm) helps to analyse results that lay partly outside the usually defined Arctic region. For correlations with the TP, the first correlation input was taken at the TP and the second in the Arctic region.

All presented correlations are significant at the confidence level 95%~~The lowest correlation level in Figures is ±0.32, representing 95% confidence levels of the correlations~~; only strong correlations $|R|$>0.5 ~~(at least 99.8% confidence level)~~ are discussed in this paper.

We used F-test to assess the significance of correlations. Autocorrelation in the time series is taken into account by using

 of averages, we used t-test assuming equal variances.

Detrending of seasonal time-series was done to ascertain that the correlations are not caused by mutual trends in input variables using following formula:

$$Y_i = X_i - (k \cdot year + b - X_{average}).$$

Therefore, linear trends for each parameter at each grid point were calculated for each season and all correlations were calculated twice – at first using the regular and secondly the detrended data. Detrending did not change general patterns of

correlations with TP, only negative correlation in the Greenland region intensified slightly. All discussed correlations of the regular data were also significant in analysis of detrended data. As we are exploring the connections that include long term climatic trends such as global warming, we present in this paper only correlations of the regular, not detrended data.

The next step in the analysis was to remove from the correlations the effect of atmospheric teleconnections which could be described by known teleconnection indices. For that purpose, partial correlations between selected atmospheric variables with

the controlling effect of the teleconnection indices were calculated as follows:

$$R_{AB|C} = \frac{R_{AB} - R_{AC} \cdot R_{BC}}{\sqrt{(1 - R_{AC}^2) \cdot (1 - R_{BC}^2)}}$$

Cold and mild winters were defined as years when the winter average temperature differed the whole period average more than one standard deviation at a geographical point in the Greenland sector (70°N, 60°W). Accordingly – cold winters were 1983, 1984, 1989, 1990, 1992 and 1993; mild winters were 1980, 1985, 1986, 2003, 2007, 2009, 2010 and 2011.

The last phase of the analysis was to calculate the lagged correlation coefficients with the purpose of revealing the possible delayed dependences between the atmospheric variables of the Arctic region and the TP. For the lagged correlation, the second parameter was taken by lag months earlier than the first parameter.

## 3 Results

### 3.1 Spatial correlations of climatic variables

Climatic variables at separate grid points are usually not independent, but correlations in space depend highly on the distance and climatic variables. For example, for temperature, the dependence in space stays significant for longer distances than for precipitation as the processes of their formation are different. But besides the short distance correlation of climatic parameters between the TP and the surrounding grid points, there are also vast areas far from the TP, still having significant correlations (Figure 2). The strongest correlations are detected in winter when temperature at the 1000 hPa level at the TP has a positive

correlation over a large area, covering nearly the whole northern Eurasia, with the maximum (R>0.5) in northern Europe, on the Eastern European Plain and Central Siberia. At the same time, an area of a strong negative correlation (R<-0.5) is found in the Greenland sector. Hereinafter, we define the Greenland sector as region between 55–80°N and 20–80°W. A similar correlation pattern, but of less magnitude, is also present in spring and autumn. The pattern of spatial correlation for temperature at the 1000 hPa level in summer is different. The area of positive correlation is much smaller than in winter, mostly

covering only Europe, but a negative correlation is detected in the central Arctic and western Siberia.

Specific humidity at the 1000 hPa level has a similar pattern of correlations as temperature at the same seasons. The largest differences are observed in Siberia in spring with about 20% higher correlation in temperature than in specific humidity. Wind speed at the 1000 hPa level at the TP has the highest correlation in winter, while the areas of a positive correlation in Europe and North Atlantic and of a negative correlation in the central Arctic and the Greenland sector are strictly distinct. During the

other seasons the spatial correlation is much lower. There is a strong positive correlation in wind speed in summer between the TP and the Canadian Arctic Archipelago and the Bering Sea region. SLP at the TP has a significant negative correlation with

SLP in the Greenland sector in autumn. Figure 2 using ERA-Interim data (not shown) gave very similar results. Detectable differences were found in Central Arctic in summer and autumn when correlations with temperature and specific humidity were slightly higher in ERA-Interim than NCEP-CFSR.

The Greenland sector showed most often significant correlations with the parameters of the Eastern Baltic Sea region. In Table 1 are given spatial average, minimum and maximum values of seasonal correlations between the TP and the Greenland sector. Strong negative correlation in the Greenland sector at 1000 hPa temperature in winter and spring decreases with altitude and turns even positive at 250 hPa (Table 1). Specific humidity at 1000 hPa shows quite similar values with temperature at the same level (Table 1). The correlation between wind speed at 1000 hPa at the TP and the Greenland sector is mostly negative in winter (Figure 2), reaching up to -0.72 (Table 1). Most significant correlation between SLP is present in autumn and summer (Figure 2, Table 1).

Climatic variables have close relationships between themselves. If there is a climatic change in one parameter, for example in temperature, then it causes changes also in other parameters connected with it, for example in ice concentration. Similarly to correlations of the same climate variable at the TP and the Arctic (Figure 2), there are strong correlations ($|R|>0.5$) between different climate variables at the TP and the Greenland sector (Table 1). Correlation between the 1000 hPa temperature at the TP and SLP show expected results with a positive correlation in summer and negative in winter around the TP (not shown). Largest correlations with the Greenland sector are in winter when the correlation is averagely –0.39 and extremal values reach even to -0.73. Correlations with ice concentration have quite large extremal values reaching up to 0.71 in summer for wind speed and 0.64 in spring for temperature, but these correlations are significant only at narrow coastal areas around Greenland (not shown).

### 3.2 Impact of the teleconnection indices

To analyse the impact of the seven teleconnection indices given at Data paragraph, the average of partial correlations between 1000 hPa temperature at TP and the Greenland sector (55–80°N, 20–80°W) are shown for winter and spring in Table 2. The influence of teleconnection indices depends strongly on a season and on a parameter. Larger difference from regular correlation values means higher impact of the index. According to the definition – removing the impact may decrease but may also increase the correlation.

The first row of Table 2 shows the average of the regular Pearson correlation of temperature at 1000 hPa in the region. It has the most significant values during winter and spring. Also the impact of AO and NAO is most considerable during these seasons. Considering the correlation coefficients between seasonal mean temperatures, specific humidity, wind speed at the 1000hPa level and SLP at the TP the AO indices have mostly higher correlations than the NAO indices, only SLP in summer and autumn has a significantly higher correlation with the NAO index (not shown). Hereafter mostly only AO index is analysed (and not NAO index).

Partial correlations with the controlling factor AO index reduce the area with a statistically significant correlation around the TP in all parameters and in all seasons. This effect on the remote areas depends on the season. In winter the effects of the AO indices on spatial correlations are the strongest, up to 0.5. In spring, the differences between partial correlations with the AO indices are below 0.2 in the whole region compared to the regular correlations between the TP and the Arctic. In summer and autumn, the differences are even smaller than in spring.

Partial correlation in temperature, removing the influence of the AO index, is below ±0.5 on all levels (1000, 850, 500 and 250 hPa) in winter, though the regular correlations are the strongest (not shown). In other seasons, regions with stronger partial correlations than ±0.5 remain. In summer and autumn, the AO indices have no significant influence on correlations on higher altitudes, similarly to the 1000 hPa level.

The impact of other teleconnection indices than AO and NAO is much smaller. Among other indices the SCA index has the strongest impact in winter but very small impact during other seasons while in spring the partial correlation with the PEU

index decreases the value of correlation coefficient the most (except AO and NAO). The average (regular) correlation coefficients between 1000 hPa temperature at TP and the Greenland sector during summer and autumn were only 0.15 and -0.02 respectively and are not discussed here.

### 3.3 Comparison of winters with low and high temperature

To compare broad atmospheric circulation patterns, we turn to the difference map of the geopotential heights of 500 hPa and temperature at 1000 hPa by subtracting the composites of cold winters (DJF) from those of mild winters (Figure 3). The large scale atmospheric circulation pattern in Figure 3 shows that the geopotential heights of 500 hPa are more than 100 gpm higher in mild winters than in cold ones. The maximum of this height anomaly is centred over the maximum of the 1000 hPa temperature difference. The whole column (up to 500 hPa) of the air in the Greenland sector is warmer than at cold years. Coming down to the lower surfaces (700 hPa, SLP, not shown), the maximum height anomaly is shifted to the east. Ensuing spring and summer also show positive values of the 1000 hPa temperature and the geopotential heights of 500 hPa in the Greenland sector (spring and summer in Figure 3). In autumn the positive anomaly of 1000 hPa temperature and geopotential heights of 500 hPa are present in Siberia.

Along the 60°W vertical slice the spring atmosphere exhibit baroclinic structure between about 60°N and 82°N due to negative height anomalies in the lower troposphere below the 850 hPa and with further higher the positive ones (spring in Figure 4). Similarly to Wu et al. (2013) the vertical distribution of spring height anomalies differs from that of the previous winter when height anomalies show dominantly quasi-barotropic structure. The annual evolution of 500 hPa height differences at 60°N shows that the positive ~~temperature~~ height anomaly at the Greenland sector shifts towards east during the next seasons, reaching to Scandinavia/Baltic Sea region in summer (Figure 5). The propagation of the mid-tropospheric anomalies in this region is nonlinear: these height anomalies are significant only over some areas and months and in May they are slightly negative. Also at 65°N the similar pattern is present (not shown), but at 70°N and 75°N this kind of signal propagation is missing. ERA-Interim has at 60°N similar patterns, but without considerable positive difference at the Greenland sector in winter (not shown).

### 3.4 Teleconnection using lagged data

~~There is a large inertia in the atmosphere causing lag effects.~~ The climate system consists of various interactive components that have highly various response times. The estimated time scales in atmosphere grow with height and reach up to months, but due to atmospheric interactions with the oceans and cryosphere, the ~~It means that climatic~~ conditions in atmosphere may have even longer response times. ~~in a previous period can have an effect on the weather during the following weeks, months and seasons.~~ For finding the effect of the previous seasons on ~~weather~~ atmospheric conditions at the TP, lagged correlations were calculated for the 1000 hPa temperature (Figure 6). The results show that the previous winter season has a strong effect on temperature during the following spring (lag=3) and summer (lag=6). At the same time, the winter mean temperature has almost no dependent on weather conditions during the previous seasons, there is only small region with strong negative correlation in the Taimyr region in the previous summer (lag=6). There is a strong ($R>0.5$) positive correlation between the 1000 hPa temperatures at the TP in spring and in Eurasia during the previous winter (lag=3). The spring temperature is not determined by the temperature of the previous autumn (lag=6) nor of the previous summer (lag=9). Summer temperature at the TP has a strong positive correlation in the Greenland sector with the previous spring (lag=3), winter (lag=6) and autumn (lag=9). Autumn temperature at the TP has a strong negative correlation with the Fram Strait in the previous summer (lag=3) and Taimyr region in the previous winter (lag=9).

### 4 Discussion

There are vast areas in the Arctic far from the Eastern Baltic Sea region that show significant correlations with meteorological

parameters at the TP. Temperature at the 1000 hPa level at the TP has a strong positive correlation (R>0.5) with the Eastern European Plain up to Central Siberia and a strong negative correlation with the Greenland sector during all seasons except summer (Figure 2). These patterns are similar to the correlations with the AO index (not shown) and are probably partly induced by the general circulation of the atmosphere. These correlations can be considered as an effect of stronger westerlies that carry relatively warm and moist air from the North Atlantic into Eurasia and, at the same time, cold and dry air from the central Arctic to Greenland and the Canadian Arctic Archipelago. As specific humidity and temperature are strongly coupled, specific humidity has a similar pattern of correlations as temperature. Warmer air can hold much more water vapour. The reason why summer season differs from other seasons maybe caused by a less effective large-scale circulation. Also, the circumstances are different in summertime: positive atmospheric energy budget, more specific humidity and clouds, the melting ice temperature is conserved by the melting energy of ice. Consequently, we assume that weather conditions in summer are more influenced by local factors, such as differences in local radiation and heat balances determined by local geographical peculiarities, and less affected by large-scale atmospheric circulation.

Partial correlation analyses were used as the control for the potential effects of different teleconnection indices on correlations between meteorological parameters at the TP and the Arctic. Partial correlations with the controlling factors of the AO and NAO indices had the strongest influence on the correlations between meteorological parameters between the TP and the Arctic region; other teleconnection indices had much smaller influence (Table 2). Budikova (2012) suggests that the AO and NAO are very closely related and the NAO is frequently referred to as a "local expression" of the AO as it dominates its structure in the Atlantic sector. Still, some scientists show different impact of NAO and AO to meteorological parameters and phenomenon. According to the review article by Bader et al. (2011), the NAO is the most important teleconnection index for investigating the impact of the Arctic sea-ice changes on teleconnection patterns. The study of Ambaum et al. (2001) suggests also that because of the physical background of the NAO index it may be physically more relevant and robust for the Northern Hemisphere variability than is the AO index. Uotila et al. (2015) preferred NAO to AO analysing Baltic Sea ice conditions because of the centre of action that has more influence over the Baltic Sea ice conditions (Uotila et al., 2015). Thompson and Wallace (1998) declared that the AO index is actually more strongly coupled to the Eurasian winter surface air temperature than the NAO index. Rinke et al. (2013) showed through the coupled regional climate model experiments that atmospheric large-scale circulation in a winter following a low September sea ice resemble a negative AO pattern. Our results show that for the correlation coefficients between the Eastern Baltic Sea region and the Greenland sector the AO index had mostly the highest impact in every season, only SLP in summer and autumn had a significantly stronger impact with the NAO index. The strongest teleconnections between the same parameter at the Eastern Baltic Sea region and the Arctic region can be found in winter, but they are clearly affected by the AO index (Table 2). ~~After the removal of the influence of the AO index, the correlations were below 0.5 in winter, while in other seasons there remained regions with strong correlations. It means that climatic teleconnections, i.e. significant spatial correlations of climatic parameters between regions located long distance from each other, in winter are mostly caused by large-scale atmospheric circulation described with the AO index.~~

Among other teleconnection indices (except AO and NAO) the SCA index showed the largest impact between 1000 hPa temperature at TP and the Greenland sector during winter (Table 2). The SCA pattern has been shown to influence precipitation, temperatures, and cyclone activity across northern Europe and Eurasia (Moore et al., 2013; Bueh and Nakamura, 2007; Seierstad et al., 2007). Uotila et al. (2015) found that the PDO index has significant impact on the Baltic Sea ice concentration although physical mechanisms linking the Baltic Sea ice with PDO are not well known (Vihma et al., 2014). We investigated the correlation between 1000 hPa temperature at TP and the Greenland sector and removing the influence of PDO (by partial correlation) did not change the results much during any season. Removing the EA/WR index influenced the results even less although regionally the anomaly centres of the EA/WR Pattern include Europe and North Atlantic. Lim (2015) analysed the EA/WR (1979–2012) and found that the positive (negative) EA/WR is associated with a strong cooling (warming) over the Ural Mountains of northern Russia which is much more eastward from our TP.

The comparison of 700 hPa and 500 hPa geopotential height differences between mild and cold winters showed that 700 hPa geopotential height is shifted to the east. This could be due to warmer sea surface of the Northern Atlantic compared to the regions that lay to west of it. The positive temperature anomaly at 1000 hPa height shifts from the Greenland sector in winter towards east reaching to Scandinavia/Baltic Sea region in summer. This could be also followed by our lagged analyses which show that the summer temperature at TP has significant correlation with Greenland sector in winter (JJA; lag=6). According to Wu et al. (2013) the summer atmospheric circulation anomalies in the northern Eurasia are associated with the previous winter SIC west of Greenland. The mechanism is based on horseshoe-like pattern of SST anomalies in the North Atlantic that persist in winter and spring. Such anomaly impacts on ensuing spring atmosphere over the North Atlantic which links winter-spring SIC and SST anomalies and summer atmospheric circulation anomalies over northern Eurasia including the Baltic Sea region. This proposed mechanism supports our results.

By Wu et al. (2013) proposed mechanism, that associates the summer atmospheric circulation anomalies in the northern Eurasia with the previous winter ice conditions west of Greenland, supports our results.

## 5 Conclusions

Rapid warming and reduction of sea ice is going on in the Arctic. In this article, the relations between meteorological parameters of the Arctic region and the Eastern Baltic Sea region are investigated, using the NCEP-CFSR and ERA-Interim reanalysis data for 1979–2015. The Eastern Baltic Sea region is characterized by meteorological values at the testing point (TP) in southern Estonia (58°N, 26°E). There are vast areas in the Arctic, far from the Eastern Baltic Sea region, that show significant correlations with climatological parameters at the TP. The most important findings about the Arctic teleconnections with the Eastern Baltic Sea region are as follows:

• The strongest teleconnections between the same parameter in the Eastern Baltic Sea region and the Arctic are in winter, but they are clearly affected by the Arctic Oscillation index (AO index). After removal of the AO index variability, correlations in winter were below 0.5, while in other seasons strong ($|R|>0.5$) correlations mostly remain.

• Strong teleconnections are present in temperature profiles from 1000 to 500 hPa. Similarly to the 1000 hPa level, teleconnections on higher levels are connected with the AO index variability in winter.

• Strong teleconnections are present between different climate variables at the TP and the Arctic. Temperature and wind speed at the 1000 hPa level in the Eastern Baltic Sea region have in all seasons strong teleconnections with the sea ice concentration in some regions of the Arctic Ocean. These teleconnections cannot be explained solely with the climate indices variability.

• The annual evolution of 500 hPa height differences (between mild and cold winter in the Greenland sector) at 60°N shows that the positive temperature anomaly during winter shifts towards east during the next seasons, reaching to Scandinavia/Baltic Sea region in summer. Also at 65°N the similar pattern is present. At higher latitudes (70 and 75°N) this kind of signal propagation is missing.

• In all seasons there are strong teleconnections in temperature at 1000 hPa at the TP with some Arctic regions from the previous seasons. The most permanent lagged correlations in 1000 hPa temperature are in summer at the TP with the Greenland sector in the previous spring, winter and even autumn.

In conclusion, at every season there are some regions in the Arctic that has strong teleconnection ($|R|>0.5$, $p<0.002$) with temperature, SLP, specific humidity, and wind speed in the Eastern Baltic Sea region. These relationships can be explained by the AO index variability only in winter. In other seasons there has to be other influencing factors. The positive temperature

anomaly evolution of 500 hPa height differences from the Greenland sector in winter to Baltic Sea region in summer is present at 60°N and 65°N meridian and missing at higher latitudes. The lagged correlation in 1000 hPa temperature in summer at the TP supports the results of temperature evolution. The results of this study are valuable for selecting regions in the Arctic that have statistically the largest effect on climate in the Eastern Baltic Sea region.

## Acknowledgements

Comments of four anonymous reviewers have led to a significant improvement of this paper. This study was supported by the Estonian Research Council grant PUT (645) and institutional research funding IUT (2-16) and IUT (20-11) of the Estonian Ministry of Education and Research. ECMWF, NCAR and NOAA-CPC are acknowledged for data supply.

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

Table 1. Areal average, minimum and maximum of seasonal correlations between VAR1 at TP and VAR2 at the Greenland sector (20–80°W, 55–80°N).

| | | AVERAGE | | | | MINIMUM | | | | MAXIMUM | | | |
|---|---|---|---|---|---|---|---|---|---|---|---|---|---|
| VAR1 | VAR2 | DJF | MAM | JJA | SON | DJF | MAM | JJA | SON | DJF | MAM | JJA | SON |
| t1000 | t1000 | -0.41 | -0.23 | 0.15 | -0.02 | -0.63 | -0.51 | -0.22 | -0.49 | 0.03 | 0.21 | 0.44 | 0.49 |
| t850 | t850 | -0.41 | -0.26 | 0.09 | -0.02 | -0.63 | -0.46 | -0.19 | -0.34 | 0.08 | 0.06 | 0.34 | 0.48 |
| t500 | t500 | -0.32 | -0.19 | 0.24 | 0.00 | -0.52 | -0.51 | 0.02 | -0.20 | 0.09 | 0.15 | 0.49 | 0.40 |
| t250 | t250 | 0.31 | 0.20 | -0.02 | 0.00 | 0.08 | 0.02 | -0.36 | -0.29 | 0.55 | 0.39 | 0.41 | 0.28 |
| q1000 | q1000 | -0.44 | -0.20 | 0.28 | -0.04 | -0.65 | -0.50 | -0.19 | -0.53 | 0.11 | 0.09 | 0.62 | 0.45 |
| s1000 | s1000 | -0.11 | 0.02 | 0.05 | 0.01 | -0.72 | -0.58 | -0.47 | -0.36 | 0.77 | 0.67 | 0.75 | 0.42 |
| SLP | SLP | 0.15 | -0.12 | -0.25 | -0.36 | -0.25 | -0.30 | -0.43 | -0.54 | 0.51 | 0.33 | 0.16 | -0.13 |
| t1000 | SLP | -0.39 | -0.27 | -0.23 | 0.03 | -0.73 | -0.50 | -0.42 | -0.45 | 0.35 | 0.11 | -0.02 | 0.51 |
| t1000 | s1000 | -0.15 | -0.03 | -0.02 | -0.03 | -0.67 | -0.54 | -0.42 | -0.65 | 0.65 | 0.48 | 0.38 | 0.43 |
| t1000 | icec | 0.17 | 0.11 | -0.07 | 0.01 | -0.26 | -0.34 | -0.62 | -0.41 | 0.61 | 0.64 | 0.41 | 0.48 |
| s1000 | icec | 0.19 | 0.06 | 0.03 | 0.16 | -0.28 | -0.67 | -0.36 | -0.24 | 0.63 | 0.63 | 0.71 | 0.59 |

Table 2. Areal average of seasonal (winter and spring) partial correlations between 1000 hPa temperature at TP and the Greenland sector (20-80°W; 55–80°N) using different teleconnection indices as controlling factors.

| index | DJF | MAM |
|---|---|---|
| reg. correl. | -0.41 | -0.23 |
| AO | -0.07 | -0.10 |
| NAO | -0.10 | -0.11 |
| PDO | -0.45 | -0.26 |
| CAI | -0.41 | -0.21 |
| PEU | -0.42 | -0.18 |
| EA | -0.43 | -0.27 |
| EA/WR | -0.41 | -0.22 |
| SCA | -0.25 | -0.23 |

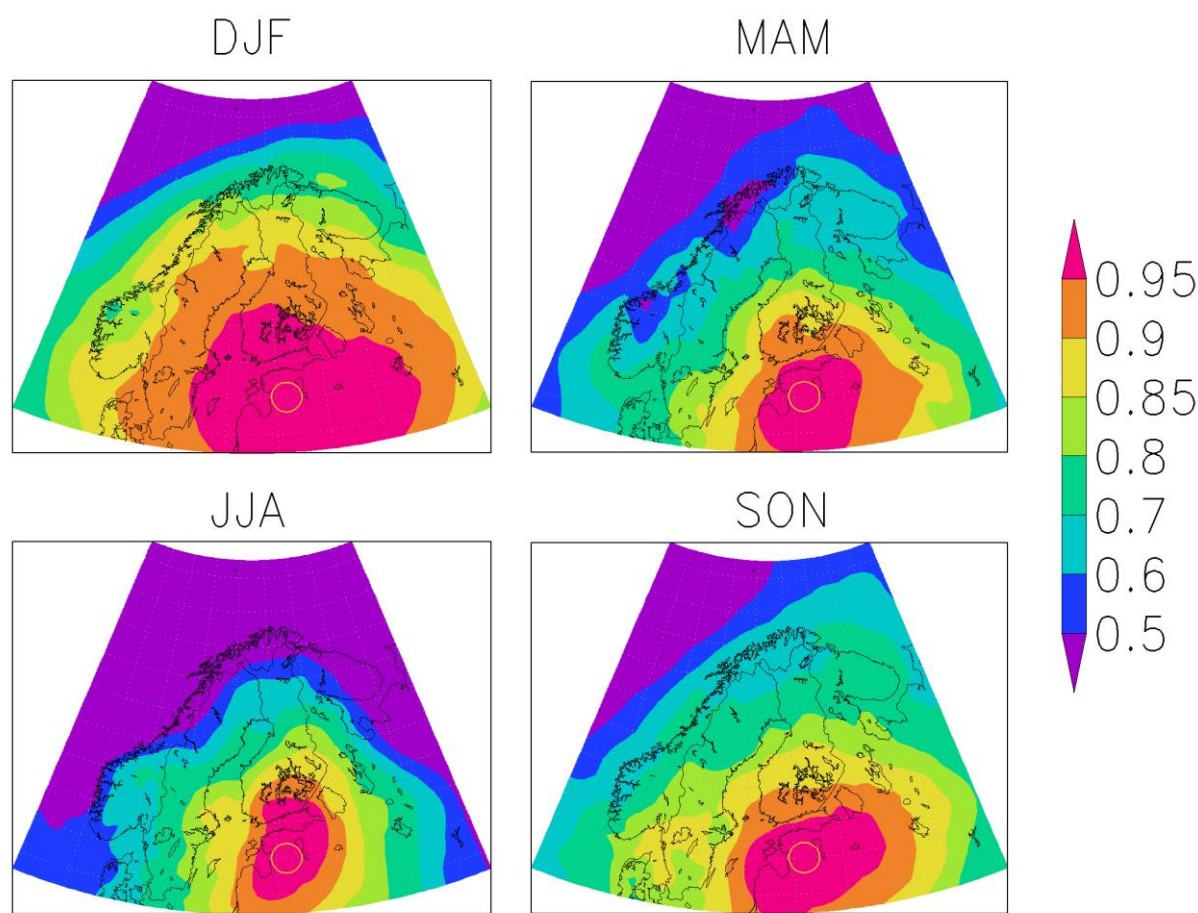

**Figure 1: Correlation maps of air temperature on 1000 hPa level for the testing point in the Baltic Sea region.**

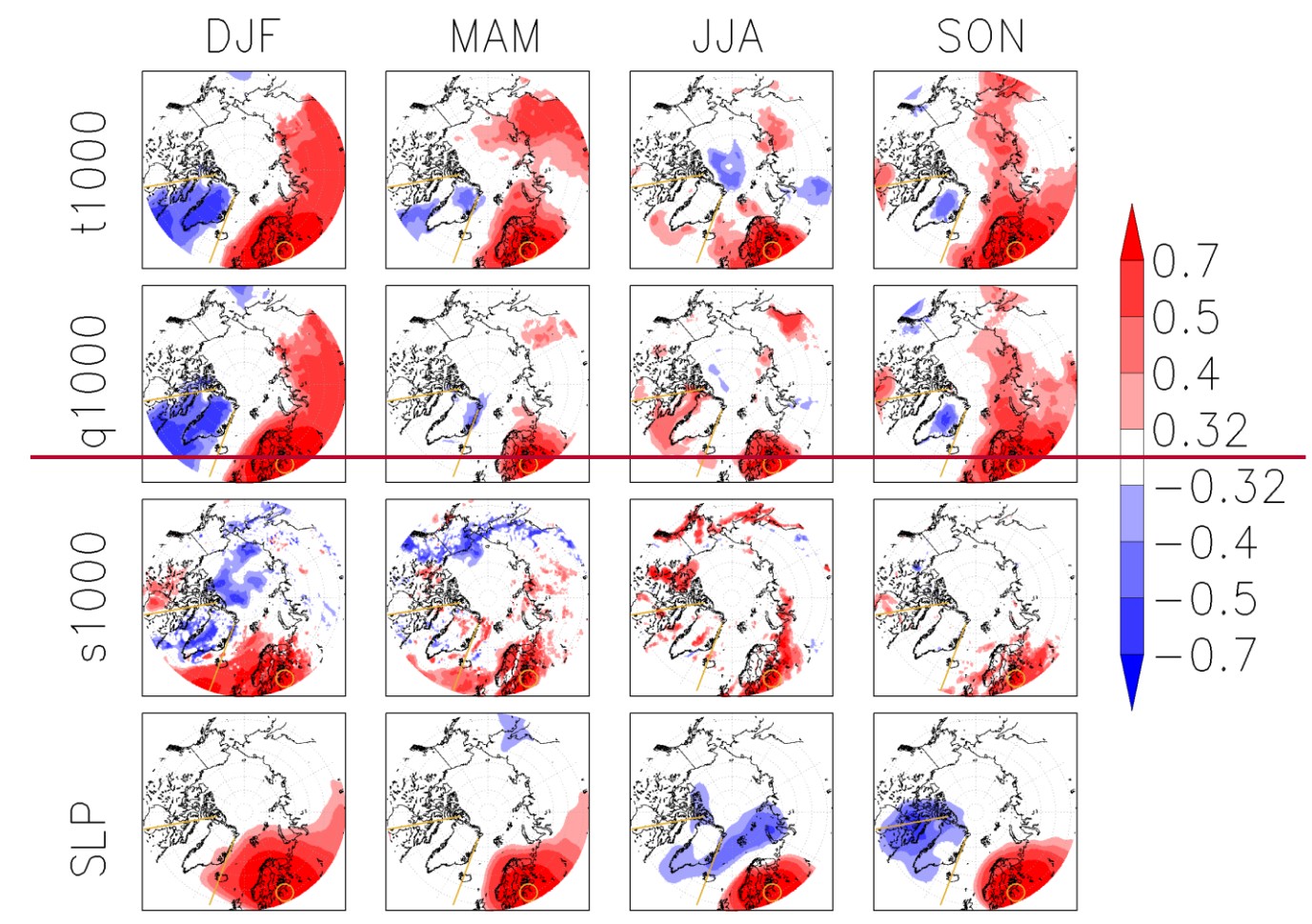

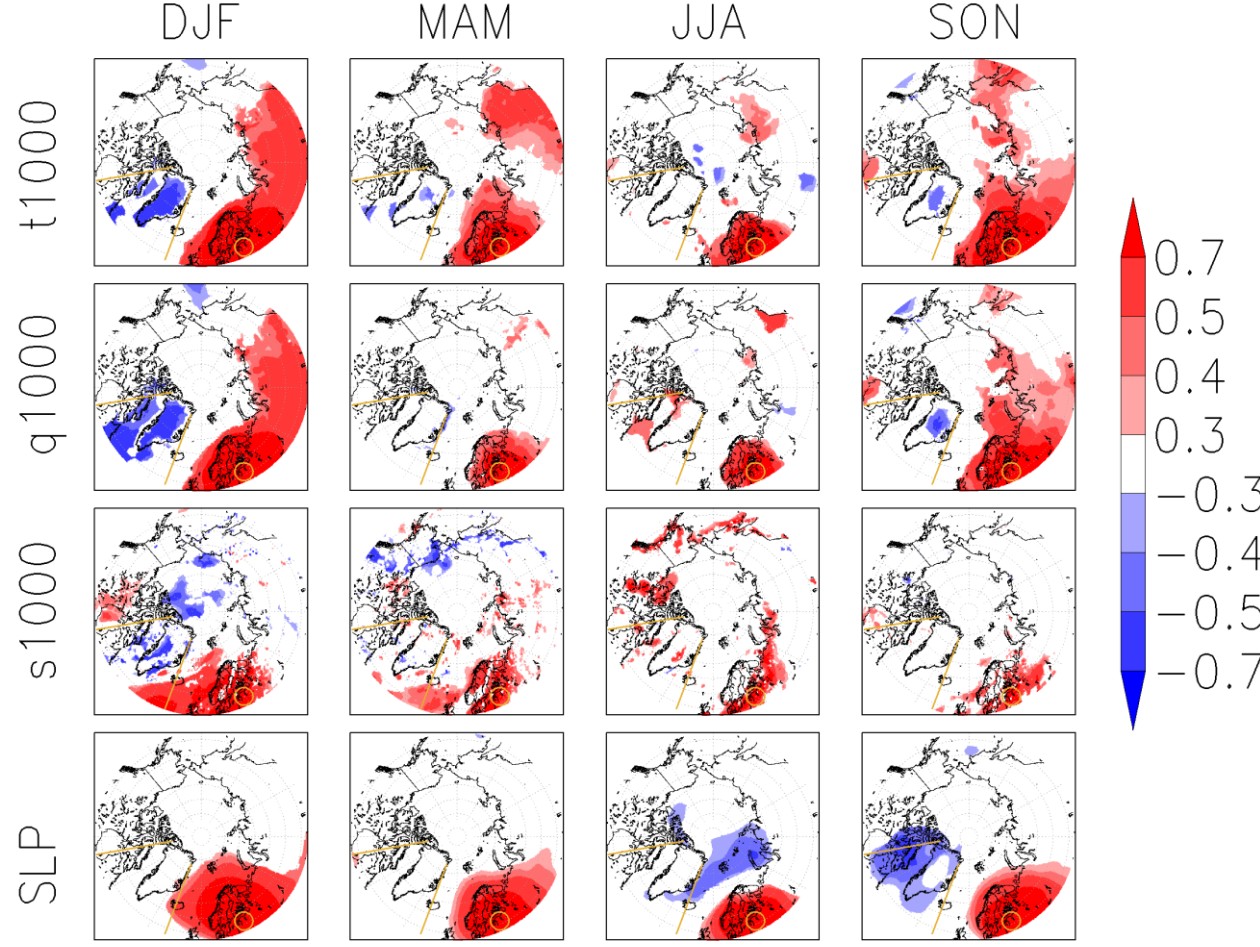

**Figure 2:** Correlation maps between seasonal mean 1000 hPa temperature (t1000), specific humidity (q1000), wind speed (s1000) and SLP measured at the TP (the yellow circle) and in the whole Arctic region. Columns represent seasons, ~~all presented correlations are shading level ±0.32 represent correlation~~ significan~~tee~~ at the confidence level 95%. The Greenland sector (20 − 80°W, 55 − 80°N) borders are marked with two yellow lines.


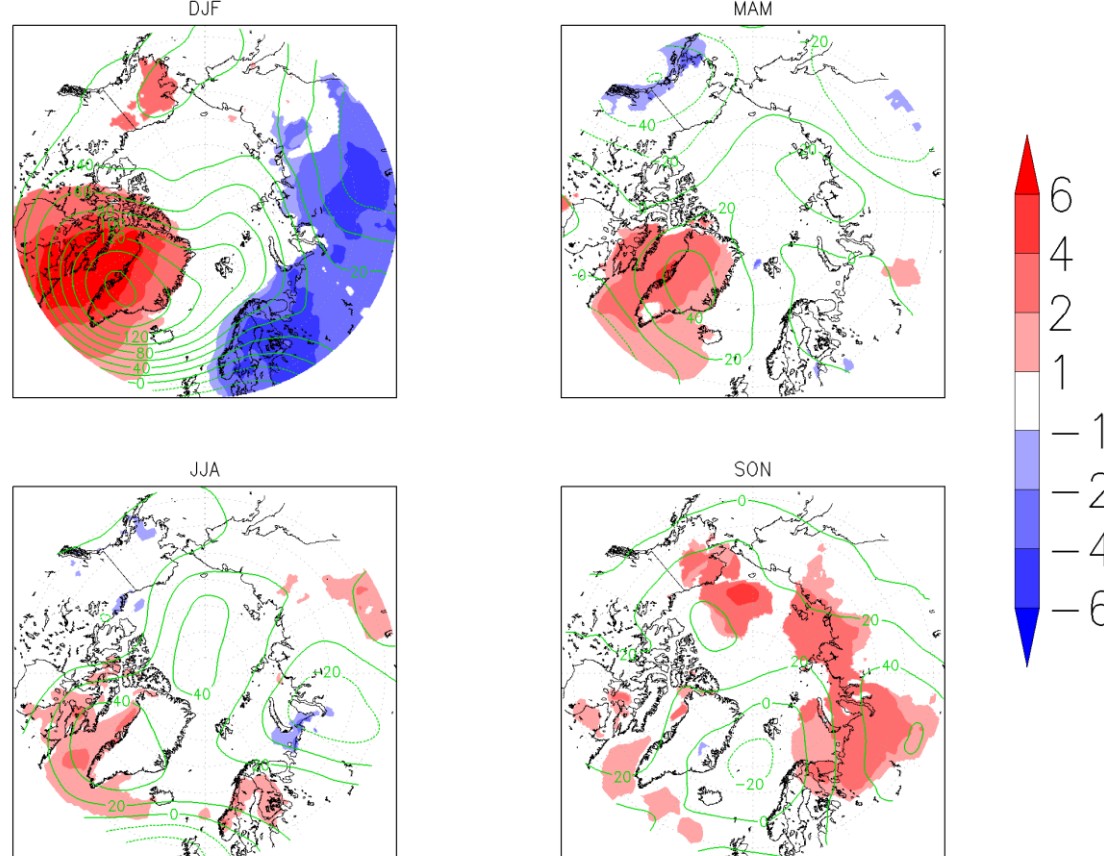

**Figure 3. Seasonal difference maps (years with mild winters minus years with cold winters) in air temperature at 1000 hPa level (shading with confidence level of 95%), and geopotential height at 500hPa level (contours).**


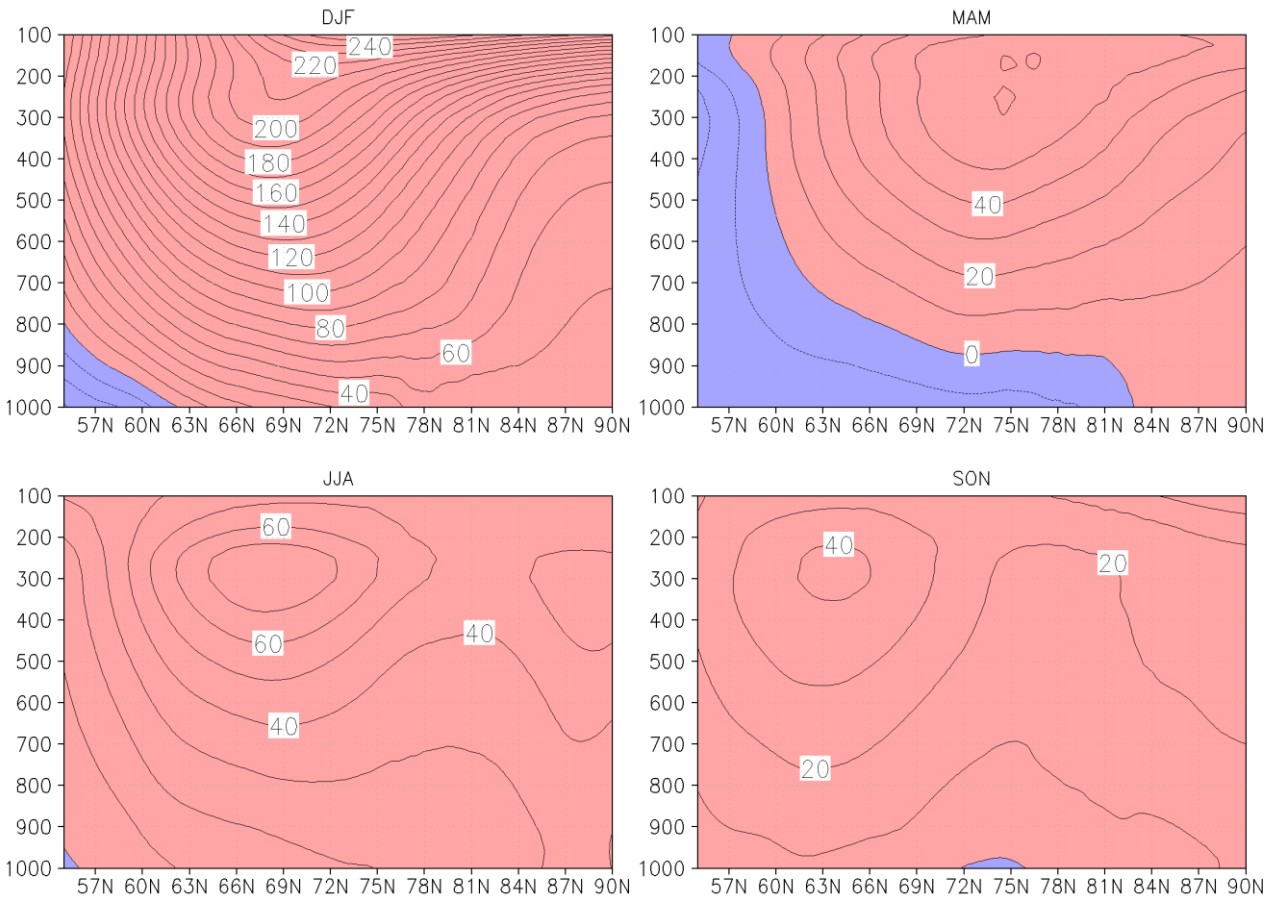

**Figure 4. Differences in the mean geopotential heights between mild and cold winters along the 60W vertical slice. Contour intervals are 10 gpm; blue represent negative height differences and red positive height differences.**

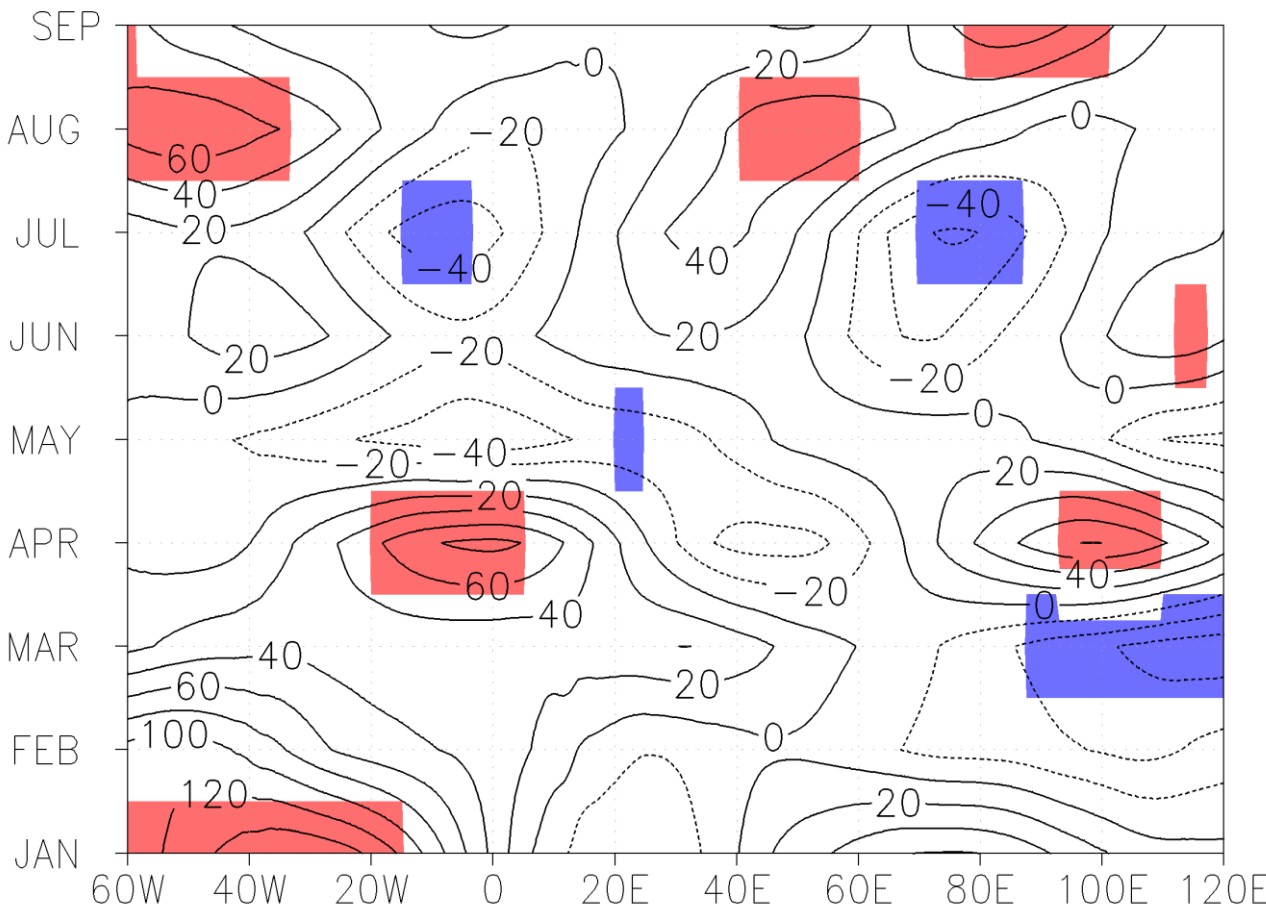


**Figure 5. Evolution of 500-hPa height differences between mild and cold winters at 60N; red and blue shading indicates differences at the 95% significance levels for positive and negative height, respectively.**


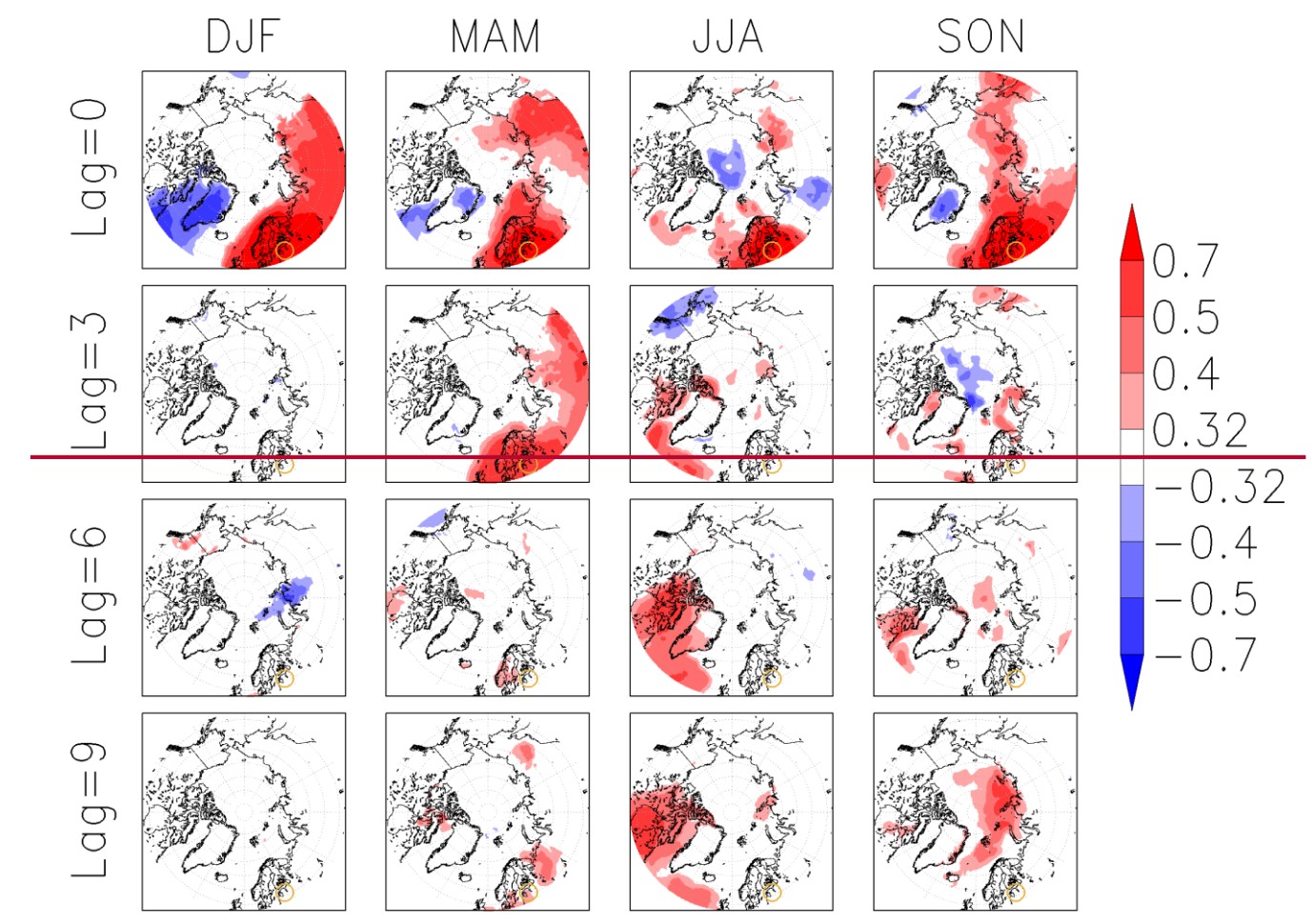

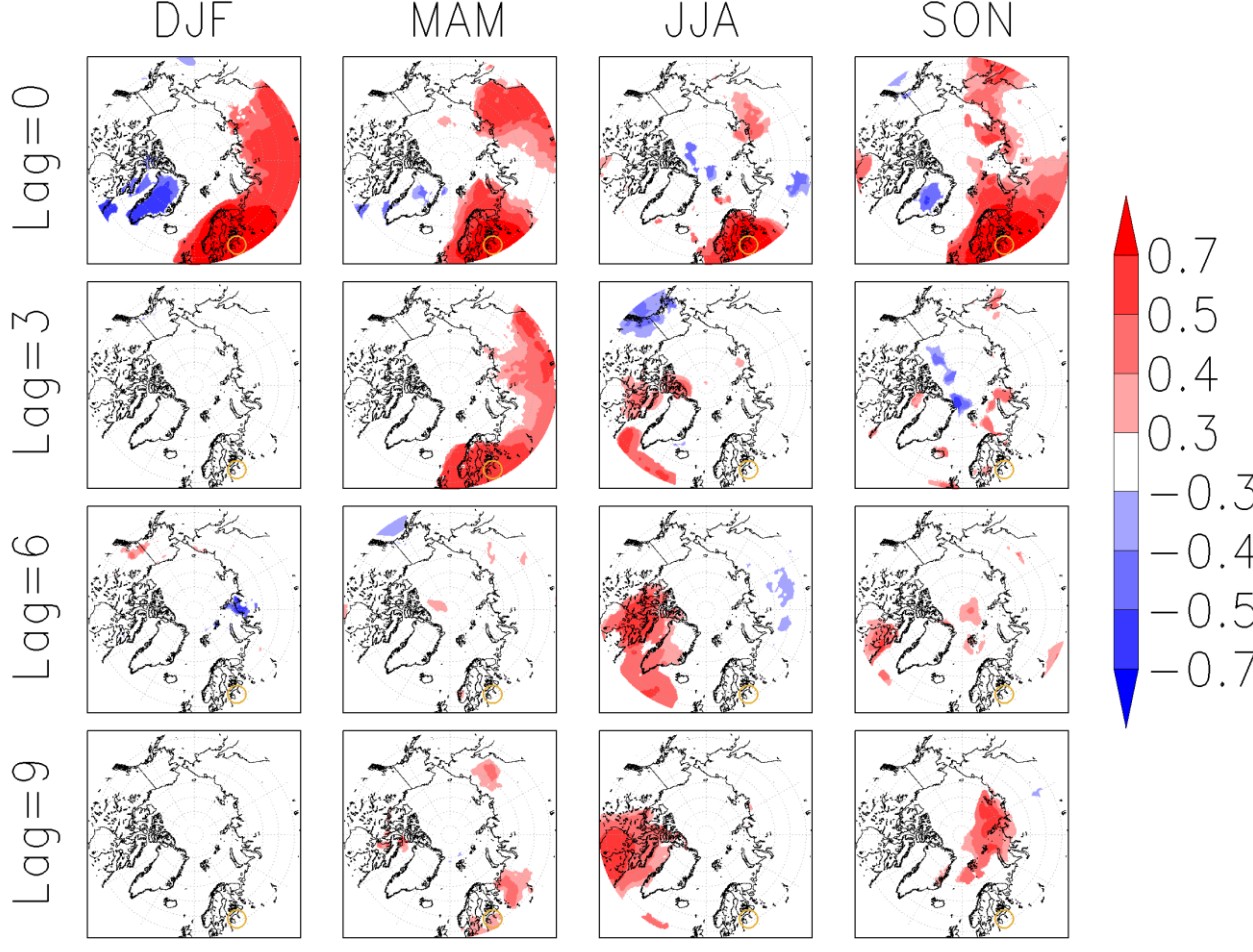

**Figure 6: Lagged correlation maps between the TP (the yellow circle) and Arctic 1000 hPa temperature: 1. row: lag is 0 months (no lag); 2. row: lag is 3 months; 3. row: lag is 6 months; 4. row: lag is 9 months. Columns represent seasons, all presented correlations are significant at the confidence level 95%. ~~shading level ±0.32 represent correlation significance at confidence level 95%.~~**