# Peer review of "Atmospheric teleconnections between the Arctic and the Eastern Baltic Sea regions"

_Earth System Dynamics, 2017_

## Referee Comment (RC1) · Anonymous Referee #1 · 27 Apr 2017

I found this manuscript generally well written and using sound statistical methods with some novel appearing concepts, related to the partial correlation applied in the context of downscaling to southern Estonian region. I have some comments and suggestions I would like the authors to address before I can recommend the publication.

Main comments:

1) More literature research should have been done to discuss on physical mechanisms behind statistically identified teleconnections. This discussion should include how AO/NAO are dependent, and therefore to some degree redundant, and the role of PDO, SCA, and PEU. For example, some studies (cf. Vihma et al. 2014; Uotila et al. 2015) have found that PDO and SCA, in addition to AO/NAO, are important for the northern Baltic Sea temperature and for the maximum Baltic sea-ice extent. This appears to contradict the author's argument that only AO/NAO are important circulation modes in the Baltic Sea region. Also, there are studies identifying possible physical mechanisms behind teleconnections. For example, Wu et al. (2013) found a linkage between the winter Baffin Bay sea-ice anomaly and northern European atmospheric circulation. Such discussion on mechanisms would assist the authors to find out which of the numerous correlation associations are likely to be physically sensible. Finally, by adding such a discussion the manuscript would better address the "Dynamics of the Earth system" subject area of the ESD journal.

2) Title is misleading. TP is a location in Southern Estonia and is not representing the entire Baltic Sea region. I base this claim on findings of previous studies mentioned above. I suggest to change the title to 'Atmospheric teleconnections between the Arctic and Southern Estonia'.

3) Results section needs to be more focussed, now the large number of details confuses the reader. As a result, the reader is left wondering which correlation links are important which are not. Here, a summary table listing the most significant and physically relevant linkages would help the reader. Such a table would then support discussion.

4) The analysis is based on only one reanalysis, although it is known that reanalyses have significant biases in the Arctic. To ensure the robustness of results, would be good to check main results with another reanalysis. I was also wondering why CFSR was picked of all available products? I suggest carrying out the analysis with an ECMWF one, such as ERA-Interim.

5) Methods have not been adequately explained. In particular, the partial correlation method needs to be explained and a reference to literature added.

6) Statistical terminology is misleading at places. For example, I would not say that correlation is strong when r is 0.5-0.7. Such a correlation range explains only 25-50% of variance.

[Figure]

7) Although the manuscript is generally clearly written, some sentences are difficult to understand (please see minor comments for details).

Minor comments:

- lines 27-28. mention what could be the mechanism linking the Arctic to the outside Arctic environment. For example, does the air advection from mid-latitudes to the Arctic change?

- line 32. 'all kinds of heat conservation changes' is obscure, be more specific.

- lines 33-34. I found this argument rather weak. So far, it has been very difficult to show that the observed Arctic warming has actually had impact on mid-latitutes.

- line 36. 'patterns of high pressure'?

- line 69. 'One of the reasons for incomplete understanding ...'

- line 71. '... vice versa due to their close proximity.'

- line 71. Where does 'Therefore' point to?

- line 100. Which correlation coefficient? Spearman?

- line 102. Why was the Arctic defined as north of 55N. Why not north of the Arctic circle?

- lines 155-160. When explaining your correlation findings, it would help if figure sub-panels are cited more frequently to specifically indicate where you see the regions. Some geographic regions mentioned are rather local and many readers may not know where they are (e.g. the Gulf of Alaska).

- line 156. 'the AO index as the controlling factor', better?

- line 171. 'change in one parameter due to climate change'?

- line 176. '... partial correlations, controlled ...'

- line 180. I can't find negative correlation in winter above Greenland and the East Siberian Sea in Figure 6.

- line 184. Positive correlation around Greenland in winter looks rather weak and not clear.

- line 188. Change to 'It means that climatic conditions'. Weather is chaotic with no memory beyond two weeks.

- line 190-191. This sentence is unclear. Do you mean '... during the following spring?'.

- line 195. How can you say that 'whole Eurasian average spring temperature is highly controlled' based on your analysis?

- line 198. You can call the region between Greenland and Svalbard the Fram Strait.

- line 209-210. I don't understand this sentence.

- line 220-221. What do you mean by 'AO/NAO paradigm'?

- line 253. 'previous season's climate conditions'.

- Table 1. Add information on the sample size, N=36?

Literature:

Uotila, P., T. Vihma and J. Haapala, Atmospheric and oceanic conditions and the extremely mild Baltic Sea ice winter 2014/15, Geophys. Res. Lett., doi:10.1002/2015GL064901, 2015.

Vihma, T., B. Cheng, and Uotila, P., Linkages between Arctic sea ice cover, large-scale atmospheric circulation, and weather and ice conditions in the Gulf of Bothnia, Baltic Sea, Advances in Polar Science, 25(4), 289-299, doi: 10.13679/j.advps.2014.4.00289, 2014.

Wu, B. Y., Zhang, R. H., D'Arrigo, R. et al., On the Relationship between winter sea ice and summer atmospheric circulation over Eurasia, Journal of Climate, 2013, 26:

5523-5536, doi:10.1175/JCLI-D-12- 00524.1.

---

## Referee Comment (RC2) · Anonymous Referee #2 · 4 May 2017

The goal of the manuscript "Atmospheric teleconnections between the greater Arctic and the Baltic Sea regions" by Jakobson and colleagues, is to explore atmospheric teleconnections between the Baltic Sea region and the greater Arctic since the late 1970s. The authors use simple statistical techniques (i.e., linear correlation analyses) of several climatic variables including air temperature, wind speeds, specific humidity, and sea level pressure. Several recognized teleconnection indices including AO, NAO, PDO, SCA, EA, and EA/WR are used in their analyses.

The scope of the work is suitable for publication in Earth System Dynamics. Additional work, however, ought to be completed before the manuscript is ready for publication. The paragraphs below include the key areas of concern as well as other minor suggestions for overall improvement of presentation.

General Comments:

1. The thrust of the paper implies that the presence of statistical correlation implies causation, which is not the case. It is important for the authors to further explore the identified relationships by placing them in a climatological context and examining various potential atmospheric processes that may help explain the correlation results.

2. The authors present a great amount of results that need to be better interpreted, synthesized and placed into a climatological/atmospheric context supported by existing literature.

3. The authors use simple linear correlation analyses to explore atmospheric teleconnections. I assume that they are speaking of the Pearson Correlation Coefficient. I have some concerns about this given that the areas of concern are in middle-to-high latitudes where teleconnections are known to be of non-linear nature. Also, the correlation method is applied to climate parameters such as wind and specific humidity that may not be normally distributed and significantly influence the results.

4. The entire Baltic Sea region is represented by one single station located in southern Estonia (TP). The authors claim that the information provided in Figure 1 (i.e., Correlations between air temperature at this location with locations across the greater Baltic Sea region during various season) shows that TP's climate represents the climate of the greater region very well. This may be the case for surface temperature, but I strongly doubt that same would hold true for the other variables such as wind characteristics. This can be seen in Figure 2 for JJA, for instance.

5. For their analyses, the authors chose four atmospheric variables including air temperature, specific humidity, wind speed, and sea level pressure. Why did they choose these variables and not just sea level pressure, or the more typical 700 hPa geopotential heights for exploring atmospheric teleconnections?

6. What methods were used to remove the trends from the data?

[Figure]

7. What methods were used to assess statistical significance?

8. The overall manuscript is clearly written baring some oddities in grammar and general use of the English language. I would recommend a more careful proof-reading of the revised manuscript. Some (not all) recommendations are included below.

Specific Comments:

Line 90: The authors mention several atmospheric teleconnections including the AO, NAO, PDO, SCA, EA, and EA/WR but do not explain what each of these are and on what basis they were included in the conversation. They also do not explain why most these were discounted up front and not addressed again even in the discussion section.

Line 105: The authors mention that they detrended the seasonal time series "to avoid the correlations to be caused by mutual trends in input variables." They also claim that the detrended and original correlation results were very similar. For this reason, they only show correlation results from "regular data". The results surprise me (i.e., similar correlations from original and detrended data), especially given the large recent temporal trends in many of the variables that are explored (i.e., temperature) in the high latitudes of the northern hemisphere. It is also important to note that the conclusions regarding teleconnections that one can reach from the original series versus detrended series may be different. Are the authors exploring the connections that include long-term climatic trends such as global warming, or are they interested in understanding the relationships as they may exist independently of such trends?

Line 191: The authors claim that ". . ., the winter mean temperature is not dependent on weather conditions during the previous seasons." But on line 199 they proceed to make the following claim: "Winter temperature at the TP has a strong negative correlation in the Taimyr region in the previous summer." To me, these statements seem to contradict themselves.

Line 235: The authors state that "To avoid false correlations, only the results that were
present in both the regular and the detrended data were discussed." I am not sure what is meant by "false correlations". Like I mentioned earlier, detrended data for instance, may hold a different story, not a false story.

Line 25: find another word for "disconfirm"

Line 26-27: It is not clear what "both" is referring to in the sentence starting with "They found that from. . .."

Line 33: "Arctic amplification" should be Arctic Amplification

Line 67: It is not customary for sentences to begin with "But"

Line 68: I would suggest replacing "last decades" with most recent?

Line 71: Rework the sentences starting with "Therefore, our aim is to. . ..

Line 123: Replace the word "huge" with large

Line 132: Can the word "distinguished" be replaced with different or distinct?

---

## Referee Comment (RC3) · Anonymous Referee #3 · 16 May 2017

The manuscript addresses the teleconnections between meteorological parameters of the Arctic and Southern Estonia. By means of correlations analyses the authors detected Arctic areas where meteorological parameters show significant correlations with southern Estonia. In winter, these statistical associations are stronger and related to the impact of the Arctic Oscillation (AO).

The Arctic key region and the dynamical linkages between the Arctic and mid- and lower latitudes is a main focus in the current climate research agenda. This study contributes to uncovering statistical relationships between recent Arctic near-surface changes in meteorological parameter and changes in NH mid-latitudes. Though the study does not contribute to the investigation of potential linkage mechanisms between the Arctic and mid- and lower latitudes, it is valuable and could be published in 'Earth System Dynamics' after major revisions addressing the following comments.

General comments

1. The background of Arctic-midlatitude linkages and possible physical relationships between Arctic climate change and midlatitude weather and climate and the role of atmospheric teleconnections has to be described more detailed and sound.

2. All analysis are based on linear correlation analysis. To make inferences about correlations, the test of the Nullhypothesis of no correlation has been performed only. I think, this need to be expanded by, at least, including non-parametric approaches not relying on normally distribution, taking into account the reduction of degrees of freedom due to autocorrelation and also by estimating the confidence intervals of the correlation coefficients. Furthermore, Wallace and Gutzler (1981) introduced a stronger criterion than that of statistical significance to make inferences about teleconnections, namely reproducibility, which should be used here, too. Furthermore, the authors have to be careful not to overstate the results of the simple correlation analyses and have to be aware that correlation does not mean causation.

3. Having in mind the position of the centers of action of the teleconnection patterns over the North-Atlantic-Eurasian region, I suggest to include the analysis of statistical relationships with the Scandinavian and East Atlantic/West Russia patterns.

4. The analysis should be extended by including other reanalysis. The authors themselves are experts in evaluating reanalysis data over the Arctic(Jakobson, E., et al, GRL, 2012). The same issue has been studied by Lindsay et al., JC, 2014). Based on these evaluations I suggest, that at least ERA-Interim should be studied for comparison.

Specific comments

(1) Check the spelling of 'Arctic Amplification' throughout the manuscript.

(2) Check the spelling of 'indices' throughout the manuscript.

(3) L57: What is meant by 'cold period'
(4) L59: 'overall warming. Over which period?

(5) L62-65: Please, give references for these statements.

(6) L69: reference Lehmann et al., 2011 is not included in the list of references.

(7) L107-108: I have some doubts, that detrending changes the correlations only slightly given only an area averaged value. I would like to see the correlation maps instead.

(8) Throughout the manuscript, do not call a correlation coefficient of 0.5 as strong, it explains only 25% of variance.

(9) L237: Though I think the results of the study are valuable, they are not very surprising nor spectacular. Please, be more cautious with your formulation.

(10) Fig.2 to 6: Do not include the shading levels below the 95% significance level.

———————————————

[Figure]

---

## Referee Comment (RC4) · Anonymous Referee #4 · 17 May 2017

The manuscript discusses statistical links between several meteorological parameters taken at one point with coordinates 58N; 26E and elsewhere north of 55N mostly using reanalysis data. The authors mostly describe results based on linear correlation analysis but provide little physical interpretations. In particular they show that in winter the teleconnections are largely explained by AO but that in the other seasons the AO/NAO teleconnections play small role. The scope of the study is suitable for Earth System Dynamics; however I also share the concerns expressed by the other reviewers that the study requires major revision before it may become publishable in the journal. In particular I agree with Reviewer 2 that the correlation results should be analysed in order to understand underlying physical processes. Also I would like the authors to be more specific regarding novel findings in the manuscript. Naively I thought that the paper by Thompson and Wallace (2000) (and several other following studies) provides

sufficient description of the teleconnections by the AO and its impacts on the climatic variables over the whole hemisphere during winter time. The same comment concerns the other seasons. For example the study by Folland et al. (2005) is a good reference regarding the influences of the summer NAO on the climate. Regarding the lagged correlations it would be nice to see how the authors connect their study to those discussing the persistence effect, such as Kolstad et al. (2015). I think the major revision addressing these points is mandatory before the study may be published in Earth System Dynamics.

Also I found the maps in the figures are too small and they are very difficult to analyse. Some of them mostly repeat each other. For example q1000 and t1000 show very similar patterns. I wonder if it is possible to reduce the number of maps.

Technical comment: the paper by Lehmann et al., 2011 is referred to in the text but not listed in the literature section.

References:

Folland CK et al (2009) The summer North Atlantic Oscillation: past, present, and future. J Clim. 22:1082–1103

Kolstad EW, Sobolowski SP, Scaife AA. 2015. Intraseasonal Persistence of European Surface Temperatures. J. Clim.28: 5365–5374.

Thompson, D. W. J., and J. M. Wallace, 2000: Annular modes in the extratropical circulation. Part I: Month-to-month variability. J. Climate, 13, 1000–1016.
* * *

---

## Author Comment (AC1) · 23 Jun 2017

**Answers to the referee 1                                23-Jun-2017**

Thank you very much for your competent and creative comments. Please find below your comments repeated again and our answers. With the help of your advices, we have prepared a new version of our manuscript.

Main comments:

1.a) More literature research should have been done to discuss on physical mechanisms behind statistically identified teleconnections. This discussion should include how AO/NAO are dependent, and therefore to some degree redundant, and the role of PDO, SCA, and PEU. For example, some studies (cf. Vihma et al. 2014; Uotila et al. 2015) have found that PDO and SCA, in addition to AO/NAO, are important for the northern Baltic Sea temperature and for the maximum Baltic sea-ice extent. This appears to contradict the author's argument that only AO/NAO are important circulation modes in the Baltic Sea region.

*To avoid the redundancy, after the first comparison with AO and NAO, we concentrate our subsequent analysis on AO only, which shows in our analysis larger impact than NAO. In discussion paragraph we added some discussion about the AO and the NAO dependency (based on Wallace, 2000; Budikova, 2012) and emphasised that although some investigations (Uotila et al., 2015; Ambaum et al., 2001; Bader et al., 2011) have been brought out that NAO is much more relevant and robust for the Northern Hemisphere variability than is the AO, still, we found like some other authors before us (Rinke et al., 2013; Balmaseda et al., 2010; Thompson and Wallace, 1998) that AO has larger impact to the teleconnections between the Artic and the mid-latitudes.*

*Our study and studies mentioned by the referee are investigating different connections. Vihma et al (2014) and Uotila et al (2015) investigated variability in the Baltic Sea region. We concentrated on the analysis of the teleconnection between the Eastern Baltic Sea region and the Arctic region. So, PDO and SCA are influencing more the Baltic Sea region variability than the Arctic and the Baltic Sea region covariability.*

*To expose the role of different teleconnection indices we reorganized the analysis of teleconnection indices as follows (based on the suggestions of our referees):*

*we explained our choices of indices based on geographical position of the centres of action of the teleconnection patterns in data paragraph (see the segment 1 beneath);*

*to show the impact of teleconnection indices we replaced the figure 4 (and left out the figure 3) with a table which contains the average of partial correlations of all relevant teleconnection indices between 1000 hPa temperature at TP and the Baffin Bay-Greenland region (20 – 80W; 55 – 80N). The region was chosen due to the results of analysis where this region showed most often significant correlation with the parameters of the Eastern Baltic Sea region. The first row shows the average of the regular Pearson correlation of 1000 hPa in the region. It has the most significant values during winter and spring. During these seasons is also the impact of AO and NAO most considerable. See the table 1 below;*

*we added to our discussion paragraph a new segment about the role of teleconnection indices, based on our analysis and literature: Uotila et al, 2015; Lim, 2015; Comas-Bru and McDermott, 2014; Vihma et al., 2014; Moore et al., 2013.*

*Segment 1 of new version:*

*"The teleconnection indices we applied in our analyses were chosen according to the possible influence due to the geographical position of the centres of action of the teleconnection patterns over the North-Atlantic-Eurasian region. The following indices were chosen: 1) The North Atlantic Oscillation (NAO), which is the dominant mode of atmospheric variability in the North Atlantic sector throughout the year (Barnston and Livezey, 1987); 2) The Arctic Oscillation (AO), which is usually defined as the first EOF of the mean sea level pressure field in the Northern Hemisphere (Ambaum et al., 2001); 3) The Scandinavian Pattern (SCA), which consists of a primary circulation centre over Scandinavia, with two other weaker centres of action with the opposite sign, one over the north eastern Atlantic and the other over central Siberia to the southwest of Lake Baikal (Bueh and Nakamura, 2007); 4) The East Atlantic Pattern (EA), which consists of a north-south dipole of anomaly centres spanning the North Atlantic from east to west (Barnston and Livezey, 1987); 5) The East Atlantic/West Russia Pattern (EA/WR), which consists of four main anomaly centres: Europe, northern China, central North Atlantic and north of the Caspian Sea; 6) The Polar/ Eurasia Pattern (PEU) consists of height anomalies over the polar region, and opposite anomalies over northern China and Mongolia.; 7) Additionally, Pacific Decadel Oscillation (PDO), which is the dominant year-round pattern of monthly North Pacific sea surface temperature (SST) variability was included. Although its geographical centres are far from the Baltic Sea region, Uotila et al (2015) found that PDO correlated significantly with the ice concentration and temperature of Baltic Sea. All indices were downloaded from the NOAA-CPC database (http://www.cpc.noaa.gov)."*

**Table 1.** *The partial correlations of teleconnection indices between 1000 hPa temperature at TP and the Baffin Bay-Greenland region (20-80W; 55 – 80). Smaller (than regular) values show higher impact of the index.*

| index | DJF | MAM | JJA | SON |
|---|---|---|---|---|
| *reg. correl.* | *-0.41* | *-0.23* | *0.15* | *-0.02* |
| **AO** | **-0.07** | **-0.10** | 0.19 | 0.08 |
| **NAO** | **-0.10** | **-0.11** | 0.23 | 0.04 |
| **PDO** | -0.45 | -0.26 | 0.06 | -0.11 |
| **CAI** | -0.41 | -0.21 | 0.15 | -0.01 |
| **PEU** | -0.42 | -0.18 | 0.19 | -0.02 |
| **EA** | -0.43 | -0.27 | 0.06 | 0 |
| **EA/WR** | -0.41 | -0.22 | 0.12 | -0.12 |
| **SCA** | -0.25 | -0.23 | 0.21 | -0.01 |

1.b) Also, there are studies identifying possible physical mechanisms behind teleconnections. For example, Wu et al. (2013) found a linkage between the winter Baffin Bay sea-ice anomaly and northern European atmospheric circulation. Such discussion on mechanisms would assist the authors to find out which of the numerous correlation associations are likely to be physically sensible. Finally, by adding such a discussion the manuscript would better address the "Dynamics of the Earth system" subject area of the ESD journal.

*We added to the Introduction paragraph the following segment to summarise possible physical mechanisms behind teleconnection (Segment 2):*

*Segment 2 of new version:*
*"The relationship between AA and weather extremes and/or persistent weather patterns in mid-latitudes are mostly explained with Arctic and North Atlantic anomalous circulation regimes, waviness and strength of jet stream (Vavrus et al., 2017; Francis and Skific, 2015; Overland et al., 2015; Barnes and Screen, 2015; Francis and Vavrus, 2015; Coumou et al., 2014; Tang et al., 2013; Petoukhov et al., 2013; Francis and Vavrus, 2012). Common supposition is that sea ice declines are primarily responsible for amplified Arctic tropospheric warming. This conjecture is central to a hypothesis in which Arctic sea ice loss forms the beginning link of a causal chain that includes weaker westerlies in mid latitudes, more persistent and amplified mid latitude waves, and more extreme weather (Perlwitz et al., 2015). On the other hand Sun et al. (2016) brought out that neither sea ice loss nor anthropogenic forcing overall yield the winter cold extremes and persistence in mid-latitudes. Arctic warming over the Barents–Kara Seas and its impacts on the mid-latitude circulations have been widely discussed (Jung et al., 2017; Dobricic et al., 2016; Semenov and Latif, 2015; Kug et al., 2015; Sato et al., 2014). Another particular regional warm core (Screen and Simmonds, 2010) is East Siberian–Chukchi Seas which is related to severe winters over North America (Kug et al., 2015; Lee et al., 2015). Screen and Simmonds (2010) brought out also the third particular regional warm core – northeast Canada and Greenland which has been less investigated. Wu et al., (2013) focused on winter SIC west of Greenland, including the Labrador Sea, Davis Strait, Baffin Bay, and Hudson Bay and found that winter SIC west of Greenland is a possible precursor for summer atmospheric circulation and rainfall anomalies over northern Eurasia. If we look at the regions in mid-latitudes then potential Arctic connections in Europe are less clear then with North America and Asia (Overland et al., 2015)."*

*To have a more focused paper we reduced the number of parameters, for that we made a general table of correlations with all our parameters and then chose only 3 for subsequent analysis: temperature, SLP and we added geopotential heights. We separated cold and warm winters (based on Baffin Bay region), similar to Sato et al, (2014); and added following analysis to reveal possible physical mechanisms why the Baltic Sea and the BB winters are in opposite phase relying on 1000 hPa temperature. We look atmospheric circulation differences using SLP, 700 hPa and 500 hPa geopotential height differences between warm and cold winters. We added also a cross-section of geopotential heights (up to 100 hPa) along the 60W vertical slice and plots of annual evolution of 500-hPa height differences at 60N, 70N and 75N (similar to Wu et al., 2013). See figures below:*

[Figure]

**Figure 1.** Seasonal difference maps (years with mild winters years with cold winters) in air temperature at 1000 hPa level (shading with confidence level of 95%), and (b) geopotential height at 500hPa level (contours).

[Figure]

**Figure 2.** Evolution of 500-hPa height differences between mild and cold winters at 60N; red and blue shading indicates differences at the 95% significance levels for positive and negative height, respectively.

[Figure]

**Figure 3.** Differences in the mean heights between mild and cold winters along the 60W vertical slice. Contour intervals are 10 gpm; blue represent negative height differences and red positive height differences.

*In discussion paragraph we added:*

*The large scale atmospheric circulation pattern in Figure 1 shows that the geopotential heights of 500 hPa are more than 100 gpm higher in mild winters than in cold ones, and the maximum of this height anomaly is centred over the maximum of the 1000 hPa temperature difference. It means that the whole column (up to 500 hPa) of the air in the Baffin Bay region is warmer than at cold years. Coming down to the lower surfaces (700 hPa, not shown), the maximum height anomaly is shifted to the east, what could be due to warmer sea surface of the Northern Atlantic compared to the regions that lay to west of it. The positive temperature anomaly (with the 500-hPa height anomalies) shifts towards east during the next seasons, reaching to Scandinavia/Baltic Sea region in summer (Figure 2). By Wu et al (2013) proposed mechanism, that associates the summer atmospheric circulation anomalies in the northern Eurasia with the previous winter ice conditions west of Greenland, supports our idea.*
*Figure 3 exhibit baroclinic structure of spring atmosphere north of 55N due to positive height anomalies in the lower troposphere below the 850 hPa and with further higher the negative ones. Similarly to Wu et al (2013) the vertical distribution of spring height anomalies differs from that of the previous winter when height anomalies show dominantly quasi-barotropic structure (not shown). With regression analysis they show the validity of their hypothesis of eastward propagation of the 500 hPa height anomalies. The same could be followed from Figure 2, where the evolution of 500 hPa height differences between mild and cold winters at 60 N is presented. Also at 65 N the similar pattern is present. At higher latitudes (70N and 75 N) this kind of signal propagation is missing.*

2) Title is misleading. TP is a location in Southern Estonia and is not representing the entire Baltic Sea region. I base this claim on findings of previous studies mentioned above. I suggest to change the title to 'Atmospheric teleconnections between the Arctic and Southern Estonia'.

*We generally agree with the reviewer. One point is not representative for the whole Baltic Sea region. But the representativeness depends very much on spatial autocorrelation of the studied parameter. To reduce the number of correlations we made a general table with all our parameters and then chose only 3 for subsequent analysis (temperature, SLP and we added height of geopotentials). According to Figure 1 in our manuscript and figure 4 in this document (see the figure 4 below) we presume that TP represents well the Eastern Baltic Sea region. Therefore we renamed the title as the 'Atmospheric teleconnections between the Arctic and the Eastern Baltic Sea regions'.*

[Figure]

Figure 4. Correlation maps of SLP for the testing point in the Eastern Baltic Sea region.

3) Results section needs to be more focussed, now the large number of details confuses the reader. As a result, the reader is left wondering which correlation links are important which are not. Here, a summary table listing the most significant and physically relevant linkages would help the reader. Such a table would then support discussion.

*To reduce the number of correlations we made a general table with all our parameters and then chose only 3 for subsequent analysis (temperature, SLP and we added height of geopotentials). We took the maximum value of the correlation between the Baffin Bay region and the testing point in the East Baltic Sea region during one season.*

4) The analysis is based on only one reanalysis, although it is known that reanalyses have significant biases in the Arctic. To ensure the robustness of results, would be good to check main results with another reanalysis. I was also wondering why CFSR was picked of all available products? I suggest carrying out the analysis with an ECMWF one, such as ERA-Interim.

*We repeated all the analysis with ERA-Interim. The results were resembled sufficiently in main points, although there were some discrepancies during summer season in the Central Arctic region. The dissimilarities are mentioned in the manuscript.*

5) Methods have not been adequately explained. In particular, the partial correlation method needs to be explained and a reference to literature added.

*We have used only well-known statistical methods in our analyses. For partial correlation, we could cite for example H. Cramer, Mathematical methods of statistics, Princeton Mathematical Series, no. 9. (Princeton University Press, Prinston, 1946), but it is in most statistical textbooks anyway. Still, we added formulas we used to the manuscript:*
*Detrending:* $Y_i = X_i - (k \cdot ye_\cdot + b - X_a \qquad )$.

*Partial correlation:* $R_{AB|C} = \dfrac{R_{AB} - R_{AC} \cdot R_{BC}}{\sqrt{(1 - R_{AC}^2) \cdot (1 - R_{BC}^2)}}$

6) Statistical terminology is misleading at places. For example, I would not say that correlation is strong when r is 0.5-0.7. Such a correlation range explains only 25-50% of variance.

*By the meaning of "how much one parameter variance is controlled by the other one", 25-50% is indeed not very strong. At the same time, by the meaning of "how certain we can be that there is connection between two parameters", the probabilities for 0.5 and 0.7 are 99.8% and 99.998% correspondingly, that is quite strong. Even correlations exceeding ±0.32 are significant at 95% confidence level, so for correlation above ±0.5 we needed stronger name than "significant".*

7) Although the manuscript is generally clearly written, some sentences are difficult to understand (please see minor comments for details).

*We improved our manuscript as suggested in minor comments:*

- lines 27-28. mention what could be the mechanism linking the Arctic to the outside Arctic environment. For example, does the air advection from mid-latitudes to the Arctic change?

*We added the assumption about southerly warm advection based on Sato et al., 2014.*

- line 32. 'all kinds of heat conservation changes' is obscure, be more specific.

*As the energy budget of the Arctic is highly dependent on energy exchange with lower latitudes, then the changes in atmospheric and oceanic circulation play an important role in all kinds of heat conservation changes in the Arctic, most prominently expressed in sea ice volume variations.*

- lines 33-34. I found this argument rather weak. So far, it has been very difficult to show that the observed Arctic warming has actually had impact on mid-latitutes.

*We added a segment to our Introduction to reveal the background of Arctic – mid latitude linkages (Segment 3) and to summarise possible physical mechanisms behind teleconnections (Segment 2).*

*Segment 3*
*"Several studies have demonstrated relationships between warming and/or ice decline, and mid-latitude weather and climate extremes (Handorf et al., 2015; Coumou et al., 2014; Tang et al., 2013; Petoukhov et al., 2013; Francis and Vavrus, 2012; Petoukhov and Semenov, 2010). Others have analysed whether these associations are statistically and/or physically robust (Hassanzadeh et al., 2014; Screen et al 2014; Barnes et al 2014; Screen and Simmonds 2013, 2014; Barnes 2013), while some investigations suggest that the apparent associations may have their origin, in part, in remote influences (Perlwitz et al., 2015; Sato et al., 2014; Peings and Magnusdottir 2014; Screen et al., 2012; Petoukhov and Semenov 2010)."*

- line 36. 'patterns of high pressure'?

*Actually we meant by the 'large-scale patterns of pressure anomalies' both high and low pressures. To be clearer we replaced the phrase as follows: 'large scale patterns of high and low pressure'.*

- line 69. 'One of the reasons for incomplete understanding ...'

*Corrected*

- line 71. '... vice versa due to their close proximity.'

*Corrected*

- line 71. Where does 'Therefore' point to?

*Therefore may-be redundant in this sentence and we decided to remove it.*

- line 100. Which correlation coefficient? Spearman?

*We use through the work only Pearson correlation, we have clarified this in the manuscript.*

- line 102. Why was the Arctic defined as north of 55N. Why not north of the Arctic circle?

*We added a sentence: "We define the Arctic region here as the region northward of 55 N. Larger region than usual (Arctic cap from polar circle or 70N; July 10 °C isotherm) helps to analyse results that lay partly outside the usually defined Arctic region."*

- lines 155-160. When explaining your correlation findings, it would help if figure subpanels are cited more frequently to specifically indicate where you see the regions. Some geographic regions mentioned are rather local and many readers may not know where they are (e.g. the Gulf of Alaska).

*We added citations of figures.*

- line 156. 'the AO index as the controlling factor', better?

*Corrected*

- line 171. 'change in one parameter due to climate change'?

*We added to the brackets (e.g. due to climate change).*

- line 176. '... partial correlations, controlled ...'

*Corrected*

- line 180. I can't find negative correlation in winter above Greenland and the East Siberian Sea in Figure 6.

*Thank you for pointing this out. There has been really some misunderstanding and the sentence is incorrect. We deleted this statement.*

- line 184. Positive correlation around Greenland in winter looks rather weak and not clear.

*It is indeed quite weak so we deleted this statement.*

- line 188. Change to 'It means that climatic conditions'. Weather is chaotic with no memory beyond two weeks.

*Corrected*

-line190-191. This sentence is unclear. Do you mean'... during the following spring?'

*We added ...'during the following spring and summer'… to clarify the sentence.*

- line 195. How can you say that 'whole Eurasian average spring temperature is highly controlled' based on your analysis?

*Our idea based on both lag=0 and lag=3 strong correlation between TP and Eurasia in spring. We tested this by changing the TP in several locations in Eurasia and it turned out that this statement is incorrect, so we deleted it.*

- line 198. You can call the region between Greenland and Svalbard the Fram Strait.

*Corrected.*

- line 209-210. I don't understand this sentence.

*We rephrase the sentence as follows:*
*The reason why summer season differs from other seasons may-be caused by a less effective large-scale circulation.*

- line 220-221. What do you mean by 'AO/NAO paradigm'?

*We rephrase the sentence as follows:*
*The study of Ambaum et al. (2001) suggests also that because of the physical background of NAO, it may be more relevant and robust for the Northern Hemisphere variability than is the AO.*

- line 253. 'previous season's climate conditions'.

*Corrected*

- Table 1. Add information on the sample size, N=36?

*Corrected*

Thank you once more for your trouble and professionalism!

Sincerely yours,

> Liisi Jakobson
> Erko Jakobson
> Piia Post
> Jaak Jaagus

**References** (If we use in our answers references that were already given in our article then we will not give the reference here again):

•       Bader, J., Mesquita, M.D.S., Hodges, K.I., Keenlyside, N., Østerhus, S., Miles, M.: A review on Northern Hemisphere sea-ice, storminess and the North Atlantic Oscillaion: Observations and projected changes, ATMOS RES, 101:809-834, 2011.

•       Balmaseda, M. A., Ferranti, L., Molteni, F. and Palmer, T. N.: Impact of 2007 and 2008 Arctic ice anomalies on the atmospheric circulation: Implications for long-range predictions, Q J ROY METEOR SOC, 136: 1655–1664. doi:10.1002/qj.661, 2010.

•       Barnes, E. A. and Screen, J. A.: The impact of Arctic warming on the midlatitude jet-stream: Can it? Has it? Will it?, WIRES CLIM CHANGE, 6: 277–286. doi:10.1002/wcc.337, 2015.

•       Barnes, E. A., Etienne, D.S., Giacomo, M., and Woollings, T.: Exploring recent trends in Northern Hemisphere blocking, GEOPHYS RES LETT, 41, doi: 10.1002/2013GL058745, 2014.

•       Barnes, Elizabeth A.: Revisiting the evidence linking Arctic Amplification to extreme weather in midlatitudes, GEOPHYS RES LETT, 40, doi:10.1002.grl.50880, 2013.

•       Barnston, A. G., and Livezey, R.E.: Classification, seasonality and persistence of low-frequency atmospheric circulation patterns, MON WEATHER REV, 115, 1083-1126, 1987.

•       Budikova, D.: Northern Hemisphere Climate Variability: Character, Forcing Mechanisms, and Significance of the North Atlantic /Arctic Oscillation, Geography Compass 6/7: 401–422, 10.1111/j.1749-8198.2012.00498.x, 2012.

•       Bueh, C. and Nakamura, H.: Scandinavian pattern and its climatic impact, Q J ROY METEOR SOC. 133: 2117 – 2131, DOI: 10.1002/qj.173, 2007.

•       Comas-Bru, L. and McDermott, F.: Impacts of the EA and SCA patterns on the European twentieth century NAO–winter climate relationship. Q J ROY METEOR SOC, 140: 354–363. doi:10.1002/qj.2158, 2014.

•       Cramer, H.: Mathematical methods of statistics, Princeton Mathematical Series, no. 9. Princeton University Press, Prinston, 1946.

•       Dobricic, S., Vignati, E., and Russo, S.: Large-Scale Atmospheric Warming in Winter and the Arctic Sea Ice Retreat, J CLIMATE, 29, 2869–2888, doi: 10.1175/JCLI-D-15-0417.1, 2016.

•       Francis, J. A. and Vavrus, S.J.: Evidence for a wavier jet stream in response to rapid Arctic warming, ENVIRON RES LETT, 10 014005, 2015.

•       Francis, J.A. and Skific N.: Evidence linking rapid Arctic warming to mid-latitude weather patterns. Philosophical transactions of the Royal Society A, 373, doi:10.1098/rsta.2014.0170, 2015.

•       Handorf, D., R. Jaiser, K. Dethloff, A. Rinke, and J. Cohen: Impacts of Arctic sea ice and continental snow cover changes on atmospheric winter teleconnections, GEOPHYS RES LETT, 42, 2367–2377. doi: 10.1002/2015GL063203, 2015.

•       Hassanzadeh, P., Kuang, Z., and B. F. Farrell: Responses of midlatitude blocks and wave amplitude to changes in the meridional temperature gradient in an idealized dry GCM, GEOPHYS RES LETT, 41, 5223–5232, doi:10.1002/2014GL060764, 2014.

•       Kug, J.S., Joeng, J.H., Jang, Y.S., Kim, B.M., Folland, C.K., Min, S.K., Son, S.W.: Two distinct influences of Arctic warming on cold winters over North America and East Asia, NAT GEOSCI, 8, 759–762, doi:10.1038/ngeo2517, 2015.

•       Lee, M.-Y., Hong, C.-C. and Hsu, H.-H.: Compounding effects of warm sea surface temperature and reduced sea ice on the extreme circulation over the extratropical North Pacific and North America during the 2013–2014 boreal winter, GEOPHYS RES LETT, 42: 1612–1618. doi: 10.1002/2014GL062956, 2015.

- Lim, Y.K.: The East Atlantic/West Russia (EA/WR) teleconnection in the North Atlantic: climate impact and relation to Rossby wave propagation, CLIM DYNAM, 44: 3211. doi:10.1007/s00382-014-2381-4, 2015.
- Moore, G.W.K., Renfrew, I.A., Pickart, R.: Multi-decadal mobility of the North Atlantic Oscillation. J CLIMATE. 26 : 2453–2466, DOI:10.1175/JCLI-D-12-00023.1, 2013.
- Overland, J., Francis, J.A., Hall, R., Hanna, E., Kim, S.J., Vihma, T.: The melting Arctic and mid-latitude weather patterns: are they connected?, J CLIMATE, . doi:10.1175/JCLI-D-14-00822.1, 2015.
- Peings, Y. and Magnusdottir, G.: Response of the wintertime northern hemisphere atmospheric circulation to current and projected arctic sea ice decline: a numerical study with CAM5, J CLIMATE, 27:244–264. doi:10.1175/JCLI-D-13-00272.1, 2014.
- Rinke, A., Dethloff, K., Dorn, W., Handorf, D., and Moore, J.C.: Simulated Arctic atmospheric feedbacks associated with late summer sea ice anomalies, J GEOPHYS RES-ATMOS, 118, 7698–7714, doi:10.1002/jgrd.50584, 2013.
- Sato, K., Inoue, J., and Watanab, M.: Influence of the Gulf Stream on the Barents Sea ice retreat and Eurasian coldness during early winter, ENVIRON RES LETT, 9, 084009, 8pp, doi:10.1088/1748-9326/9/8/084009, 2014.
- Screen, J. A., and I. Simmonds (2013), Exploring links between Arctic amplification and mid-latitude weather, GEOPHYS RES LETT, 40, 959–964, doi:10.1002/grl.50174, 2013.
- Screen, J.A. and Simmonds, I.: Amplified mid-latitude planetary waves favour particular regional weather extremes, NAT CLIM CHANGE, 4, 704-709, 2014.
- Screen, J.A. and Simmonds, I.: Increasing fall winter energy loss from the Arctic Ocean and its role in Arctic temperature amplification, GEOPHYS RES LETT, 37, L16707, doi:10.1029/2010GL044136, 2010.
- Screen, J.A., Deser, C., and Simmonds, I.: Local and remote controls on observed Arctic warming, GEOPHYS RES LETT, 39, L10709, doi:10.1029/2012GL051598, 2012.
- Screen, J.A., Deser, C., Simmonds, I., and Tomas, R.: Atmospheric impacts of Arctic sea-ice loss, 1979-2009: Separating forced change from atmospheric internal variability, CLIM DYNAM, 43, 333-344, 2014.
- Semenov, V. A. and Latif, M.: Nonlinear winter atmospheric circulation response to Arctic sea ice concentration anomalies for different periods during 1966–2012, ENVIRON RES LETT, 10 (5). 054020. DOI 10.1088/1748-9326/10/5/054020, 2015.
- Sun, L., Perlwitz, J., and Hoerling, M.: What caused the recent "Warm Arctic, Cold Continents" trend pattern in winter temperatures?, GEOPHYS RES LETT, 43, 5345–5352, doi:10.1002/2016GL069024, 2016.
- Uotila, P., Vihma, T., and Haapala, J.: Atmospheric and oceanic conditions and the extremely mild Baltic Sea ice winter 2014/15, GEOPHYS RES LETT, doi:10.1002/2015GL064901, 2015.
- Vihma, T., Cheng, B., and Uotila, P.: Linkages between Arctic sea ice cover, large-scale atmospheric circulation, and weather and ice conditions in the Gulf of Bothnia, Baltic Sea, Advances in Polar Science, 25(4), 289-299, doi: 10.13679/j.advps.2014.4.00289, 2014.
- Wallace, J. M.: North atlantic oscillatiodannular mode: Two paradigms—one phenomenon, Q J ROY METEOR SOC, 126, 564, 791-805, DOI: 10.1002/qj.49712656402, 2000.
- Wu, B. Y., Zhang, R. H., D'Arrigo, R. et al.: On the Relationship between winter sea ice and summer atmospheric circulation over Eurasia, J CLIMATE, 26:5523-5536, doi:10.1175/JCLI-D-12- 00524.1, 2013.

---

## Author Comment (AC3) · 23 Jun 2017

Thank you very much for time dedicated to our manuscript. Please find below your comments repeated again, and our answers. With the help of your advices, we have prepared a restructured, rebalanced and easier readable version of our manuscript.

General comments:

1. The background of Arctic-midlatitude linkages and possible physical relationships between Arctic climate change and midlatitude weather and climate and the role of atmospheric teleconnections has to be described more detailed and sound.

*We added the following segment to summarise possible physical mechanisms behind teleconnection and to reveal the background of Arctic - mid latitude linkages to our Introducion (the first sentence was also in the previous version):*

*"Several studies have demonstrated relationships between warming and/or ice decline, and mid-latitude weather and climate extremes (Handorf et al., 2015; Coumou et al., 2014; Tang et al., 2013; Petoukhov et al., 2013; Francis and Vavrus, 2012; Petoukhov and Semenov, 2010). Others have analysed whether these associations are statistically and/or physically robust (Hassanzadeh et al., 2014; Screen et al 2014; Barnes et al 2014; Screen and Simmonds 2013, 2014; Barnes 2013), while some investigations suggest that the apparent associations may have their origin, in part, in remote influences (Perlwitz et al., 2015; Sato et al., 2014; Peings and Magnusdottir 2014; Screen et al., 2012; Petoukhov and Semenov 2010)."*
*The relationship between AA and weather extremes and/or persistent weather patterns in mid-latitudes are mostly explained with Arctic and North Atlantic anomalous circulation regimes, waviness and strength of jet stream (Vavrus et al., 2017; Francis and Skific, 2015; Overland et al., 2015; Barnes and Screen, 2015; Francis and Vavrus, 2015; Coumou et al., 2014; Tang et al., 2013; Petoukhov et al., 2013; Francis and Vavrus, 2012). Common supposition is that sea ice declines are primarily responsible for amplified Arctic tropospheric warming. This conjecture is central to a hypothesis in which Arctic sea ice loss forms the beginning link of a causal chain that includes weaker westerlies in mid latitudes, more persistent and amplified mid latitude waves, and more extreme weather (Perlwitz et al., 2015). On the other hand Sun et al. (2016) brought out that neither sea ice loss nor anthropogenic forcing overall yield the winter cold extremes and persistence in mid-latitudes. Arctic warming over the Barents–Kara Seas and its impacts on the mid-latitude circulations have been widely discussed (Dobricic et al., 2016; Semenov and Latif, 2015; Kug et al., 2015; Sato et al., 2014). Another particular regional warm core (Screen and Simmonds, 2010) is East Siberian–Chukchi Seas which is related to severe winters over North America (Kug et al., 2015; Lee et al., 2015). Screen and Simmonds (2010) brought out also the third particular regional warm core – northeast Canada and Greenland which has been less investigated. Wu et al., (2013) focused on winter SIC west of Greenland, including the Labrador Sea, Davis Strait, Baffin Bay, and Hudson Bay and found that winter SIC west of Greenland is a possible precursor for summer atmospheric circulation and rainfall anomalies over northern Eurasia. If we look at the regions in mid-*

*latitudes then potential Arctic connections in Europe are less clear then with North America and Asia (Overland et al., 2015).*

2. All analysis are based on linear correlation analysis. To make inferences about correlations, the test of the Nullhypothesis of no correlation has been performed only. I think, this need to be expanded by, at least, including non-parametric approaches not relying on normally distribution, taking into account the reduction of degrees of freedom due to autocorrelation and also by estimating the confidence intervals of the correlation coefficients. Furthermore, Wallace and Gutzler (1981) introduced a stronger criterion than that of statistical significance to make inferences about teleconnections, namely reproducibility, which should be used here, too. Furthermore, the authors have to be careful not to overstate the results of the simple correlation analyses and have to be aware that correlation does not mean causation.

*We agree, that our real data may not fulfil all condition thoroughly that are preconditions for linear correlations, especially linearity and normality. Still, as we are seeking not exact numbers but rather general patters, small violation of normal distribution assumptions should not have considerable effect. Also – as we use mostly seasonal mean values – central limit theorem also gives us credit to assume that the data is at least in some extent normally distributed.*

*We agree, that correlation does not mean causation always as covariability between two different data can happen without reason. Still, we discuss in the paper only correlations that are larger than ±0.5. For two random 37-elements dataset there is possibility to have such big correlation 0.2%, so only 2 cases of 1000.*

*For reproducibility, we run over all calculations using ERA-interim data, and saw that these two models show similar correlation patterns. We believe that our results using linear correlations are sufficiently confidential for our conclusions even without suggested Wallace and Gutzler (1981) method that would demand totally new calculations.*

3. Having in mind the position of the centers of action of the teleconnection patterns over the North-Atlantic-Eurasian region, I suggest to include the analysis of statistical relationships with the Scandinavian and East Atlantic/West Russia patterns.

*We added explanations (also for Scandinavian and East Atlantic/West Russia patterns) our choices of indices based on geographical position of the centres of action of the teleconnection patterns in data paragraph (see the segment 1 beneath);*

*Segment 1:*
*"The teleconnection indices we applied in our analyses were chosen according to the possible influence due to the geographical position of the centres of action of the teleconnection patterns over the North-Atlantic-Eurasian region. The following indices were chosen: 1) The North Atlantic Oscillation (NAO), which is the dominant mode of atmospheric variability in the North Atlantic sector throughout the year (Barnston and Livezey, 1987); 2) The Arctic Oscillation (AO), which is usually defined as the first EOF of the mean sea level pressure field in the Northern Hemisphere (Ambaum et al., 2001); 3) The Scandinavian Pattern (SCA), which consists of a primary circulation centre over Scandinavia, with two other weaker centres of action with the opposite sign, one over the north eastern Atlantic and the other over central Siberia to the southwest of Lake Baikal (Bueh and Nakamura, 2007); 4) The East Atlantic Pattern (EA), which consists of a north-south dipole of anomaly centres spanning the North Atlantic from east to west (Barnston and Livezey, 1987); 5) The East Atlantic/West*

*Russia Pattern (EA/WR), which consists of four main anomaly centres: Europe, northern China, central North Atlantic and north of the Caspian Sea; 6) The Polar/ Eurasia Pattern (PEU) consists of height anomalies over the polar region, and opposite anomalies over northern China and Mongolia.; 7) Additionally, Pacific Decadel Oscillation (PDO), which is the dominant year-round pattern of monthly North Pacific sea surface temperature (SST) variability was included. Although its geographical centres are far from the Baltic Sea region, Uotila et al (2015) found that PDO correlated significantly with the ice concentration and temperature of Baltic Sea. All indices were downloaded from the NOAA-CPC database (http://www.cpc.noaa.gov)."*

4. The analysis should be extended by including other reanalysis. The authors themselves are experts in evaluating reanalysis data over the Arctic (Jakobson, E., et al, GRL, 2012). The same issue has been studied by Lindsay et al., JC, 2014). Based on these evaluations I suggest, that at least ERA-Interim should be studied for comparison.

*We repeated the analysis with ERA-Interim. The results were resembled sufficiently in main points in the study region, although there we some discrepancies during summer season in the Central Arctic region. The dissimilarities are mentioned in the manuscript.*

Specific comments:
(1) Check the spelling of 'Arctic Amplification' throughout the manuscript.

*Corrected*

(2) Check the spelling of 'indices' throughout the manuscript.

*Corrected*

(3) L57: What is meant by 'cold period'

*We added the explanation to the brackets: (NDJFM).*

(4) L59: 'overall warming. Over which period?

*We added a period and a reference as follows:*

*After 1980s there has been significant temperature increase in the Baltic Sea region (BACC II, 2015).*

(5) L62-65: Please, give references for these statements.

*We added the reference (BACC II, 2015).*

(6) L69: reference Lehmann et al., 2011 is not included in the list of references.

*Corrected*

(7) L107-108: I have some doubts, that detrending changes the correlations only slightly given only an area averaged value. I would like to see the correlation maps instead.

*Here are temperature at 1000 hPa seasonal correlations: first without detrending, second after detrending and third is first figure minus second one. Without trends in temperature, negative correlations with TP would be slightly stronger in Greenland-Labrador area. Our sentence "differences between the areal averages of correlations were up to 0.02 in both directions" is indeed a bit misleading, we replaced it with "detrending did not change general patterns of correlations with TP, only intensified negative correlation in the Greenland region ".*

*In the upgraded version, we show only results without detrending, to focus connections that are present in our world that is influenced by global climate change trends.*

[Figure]

[Figure]

(8) Throughout the manuscript, do not call a correlation coefficient of 0.5 as strong, it explains only 25% of variance.

*By the meaning of "how much one parameter variance is controlled by the other one", 25-50% is indeed not very strong. At the same time, by the meaning of "how certain we can be that there is connection between two parameters", the probabilities for 0.5 and 0.7 are 99.8% and 99.998% correspondingly, that is quite strong. Even correlations exceeding ±0.32 are significant at 95% confidence level, so for correlation above ±0.5 we needed stronger name than "significant".*

(9) L237: Though I think the results of the study are valuable, they are not very surprising nor spectacular. Please, be more cautious with your formulation.

*We tried to be more cautious with our formulations and replaced "spectacular" with "important".*

(10) Fig.2 to 6: Do not include the shading levels below the 95% significance level.

*We initially added 68% shading level to mark regions with still quite high possibility for the connections. We upgraded the new version to have first shading level at 95% as you suggested.*

Thank you once more for your trouble!

Sincerely yours,

Liisi Jakobson
Erko Jakobson
Piia Post
Jaak Jaagus

**References** (If we use in our answers references that are already given in our article then we will not give the reference here again):

• Barnes, E. A. and Screen, J. A.: The impact of Arctic warming on the midlatitude jet-stream: Can it? Has it? Will it?, WIRES CLIM CHANGE, 6: 277–286. doi:10.1002/wcc.337, 2015.

• Barnes, E. A., Etienne, D.S., Giacomo, M., and Woollings, T.: Exploring recent trends in Northern Hemisphere blocking, GEOPHYS RES LETT, 41, doi: 10.1002/2013GL058745, 2014.

• Barnes, Elizabeth A.: Revisiting the evidence linking Arctic Amplification to extreme weather in midlatitudes, GEOPHYS RES LETT, 40, doi:10.1002.grl.50880, 2013.

• Barnston, A. G., and Livezey, R.E.: Classification, seasonality and persistence of low-frequency atmospheric circulation patterns, MON WEATHER REV, 115, 1083-1126, 1987.

• Bueh, C. and Nakamura, H.: Scandinavian pattern and its climatic impact, Q J ROY METEOR SOC. 133: 2117 – 2131, DOI: 10.1002/qj.173, 2007.

• Dobricic, S., Vignati, E., and Russo, S.: Large-Scale Atmospheric Warming in Winter and the Arctic Sea Ice Retreat, J CLIMATE, 29, 2869–2888, doi: 10.1175/JCLI-D-15-0417.1, 2016.

• Francis, J. A. and Vavrus, S.J.: Evidence for a wavier jet stream in response to rapid Arctic warming, ENVIRON RES LETT, 10 014005, 2015.

• Francis, J.A. and Skific N.: Evidence linking rapid Arctic warming to mid-latitude weather patterns. Philosophical transactions of the Royal Society A, 373, doi:10.1098/rsta.2014.0170, 2015.

• Handorf, D., R. Jaiser, K. Dethloff, A. Rinke, and J. Cohen: Impacts of Arctic sea ice and continental snow cover changes on atmospheric winter teleconnections, GEOPHYS RES LETT, 42, 2367–2377. doi: 10.1002/2015GL063203, 2015.

• Hassanzadeh, P., Kuang, Z., and B. F. Farrell: Responses of midlatitude blocks and wave amplitude to changes in the meridional temperature gradient in an idealized dry GCM, GEOPHYS RES LETT, 41, 5223–5232, doi:10.1002/2014GL060764, 2014.

• Kug, J.S., Joeng, J.H., Jang, Y.S., Kim, B.M., Folland, C.K., Min, S.K., Son, S.W.: Two distinct influences of Arctic warming on cold winters over North America and East Asia, NAT GEOSCI, 8, 759–762, doi:10.1038/ngeo2517, 2015.

• Lee, M.-Y., Hong, C.-C. and Hsu, H.-H.: Compounding effects of warm sea surface temperature and reduced sea ice on the extreme circulation over the extratropical North Pacific and North America during the 2013–2014 boreal winter, GEOPHYS RES LETT, 42: 1612–1618. doi: 10.1002/2014GL062956, 2015.

• Overland, J., Francis, J.A., Hall, R., Hanna, E., Kim, S.J., Vihma, T.: The melting Arctic and mid-latitude weather patterns: are they connected?, J CLIMATE, . doi:10.1175/JCLI-D-14-00822.1, 2015.

• Peings, Y. and Magnusdottir, G.: Response of the wintertime northern hemisphere atmospheric circulation to current and projected arctic sea ice decline: a numerical study with CAM5, J CLIMATE, 27:244–264. doi:10.1175/JCLI-D-13-00272.1, 2014.

• Sato, K., Inoue, J., and Watanab, M.: Influence of the Gulf Stream on the Barents Sea ice retreat and Eurasian coldness during early winter, ENVIRON RES LETT, 9, 084009, 8pp, doi:10.1088/1748-9326/9/8/084009, 2014.

• Screen, J. A., and I. Simmonds (2013), Exploring links between Arctic amplification and mid-latitude weather, GEOPHYS RES LETT, 40, 959–964, doi:10.1002/grl.50174, 2013.

• Screen, J.A. and Simmonds, I.: Amplified mid-latitude planetary waves favour particular regional weather extremes, NAT CLIM CHANGE, 4, 704-709, 2014.

•	Screen, J.A., Deser, C., and Simmonds, I.: Local and remote controls on observed Arctic warming, GEOPHYS RES LETT, 39, L10709, doi:10.1029/2012GL051598, 2012.

•	Screen, J.A., Deser, C., Simmonds, I., and Tomas, R.: Atmospheric impacts of Arctic sea-ice loss, 1979-2009: Separating forced change from atmospheric internal variability, CLIM DYNAM, 43, 333-344, 2014.

•	Semenov, V. A. and Latif, M.: Nonlinear winter atmospheric circulation response to Arctic sea ice concentration anomalies for different periods during 1966–2012, ENVIRON RES LETT, 10 (5). 054020. DOI 10.1088/1748-9326/10/5/054020, 2015.

•	Sun, L., Perlwitz, J., and Hoerling, M.: What caused the recent "Warm Arctic, Cold Continents" trend pattern in winter temperatures?, GEOPHYS RES LETT, 43, 5345–5352, doi:10.1002/2016GL069024, 2016.

•	Uotila, P., Vihma, T., and Haapala, J.: Atmospheric and oceanic conditions and the extremely mild Baltic Sea ice winter 2014/15, GEOPHYS RES LETT, doi:10.1002/2015GL064901, 2015.

•	Wallace, J.M. and Gutzler, D.S.: Teleconnections in the Geopotential Height Field during the Northern Hemisphere Winter, Monthly Weather Review, 109, 784-812, https://doi.org/10.1175/1520-0493(1981)109<0784:TITGHF>2.0.CO;2, 1981.

•	Wu, B. Y., Zhang, R. H., D'Arrigo, R. et al.: On the Relationship between winter sea ice and summer atmospheric circulation over Eurasia, J CLIMATE, 26:5523-5536, doi:10.1175/JCLI-D-12- 00524.1, 2013.

---

## Author Comment (AC4) · 23 Jun 2017

Thank you very much for your comments. Please find below your comments repeated again and our answers. With the help of your advices, we have prepared a new version of our manuscript.

a) The authors mostly describe results based on linear correlation analysis but provide little physical interpretations.

*To have a more focused paper we reduced the number of parameters, for that we made a general table of correlations with all our parameters and then chose only 3 for subsequent analysis: temperature, SLP and we added geopotential heights. We separated cold and warm winters (based on Baffin Bay region), similar to Sato et al, (2014); and added following analysis to reveal possible physical mechanisms why the Baltic Sea and the BB winters are in opposite phase relying on 1000 hPa temperature. We look atmospheric circulation differences using SLP, 700 hPa and 500 hPa geopotential height differences between warm and cold winters. We added also a cross-section of geopotential heights (up to 100 hPa) along the 60W vertical slice and plots of annual evolution of 500-hPa height differences at 60N, 70N and 75N (similar to Wu et al., 2013). See figures below:*

[Figure]

*Figure 1. Seasonal difference maps (years with mild winters years with cold winters) in air temperature at 1000 hPa level (shading with confidence level of 95%), and (b) geopotential height at 500hPa level (contours).*

[Figure]

*Figure 2. Evolution of 500-hPa height differences between mild and cold winters at 60N; red and blue shading indicates differences at the 95% significance levels for positive and negative height, respectively.*

[Figure]

*Figure 3. Differences in the mean heights between mild and cold winters along the 60W vertical slice. Contour intervals are 10 gpm; blue represent negative height differences and red positive height differences.*

*In discussion paragraph we added:*

*The large scale atmospheric circulation pattern in Figure 1 shows that the geopotential heights of 500 hPa are more than 100 gpm higher in mild winters than in cold ones, and the maximum of this height anomaly is centred over the maximum of the 1000 hPa temperature difference. It means that the whole column (up to 500 hPa) of the air in the Baffin Bay region is warmer than at cold years. Coming down to the lower surfaces (700 hPa, not shown), the maximum height anomaly is shifted to the east, what could be due to warmer sea surface of the Northern Atlantic compared to the regions that lay to west of it. The positive temperature anomaly (with the 500-hPa height anomalies) shifts towards east during the next seasons, reaching to Scandinavia/Baltic Sea region in summer (Figure 2). By Wu et al (2013) proposed mechanism, that associates the summer atmospheric circulation anomalies in the northern Eurasia with the previous winter ice conditions west of Greenland, supports our idea.*
*Figure 3 exhibit baroclinic structure of spring atmosphere north of 55N due to positive height anomalies in the lower troposphere below the 850 hPa and with further higher the negative ones. Similarly to Wu et al (2013) the vertical distribution of spring height anomalies differs from that of the previous winter when height anomalies show dominantly quasi-barotropic structure (not shown). With regression analysis they show the validity of their hypothesis of eastward propagation of the 500 hPa height anomalies. The same could be followed from Figure 2, where the evolution of 500 hPa height differences between mild and cold winters at 60 N is presented. Also at 65 N the similar pattern is present. At higher latitudes (70N and 75 N) this kind of signal propagation is missing.*

b) Also I would like the authors to be more specific regarding novel findings in the manuscript.

*We rewrote our Conclusions and besides results that assure (e.g. the strongest teleconnections are present in winter; temperature has strong negative correlation) or contradict (e.g. which has more impact NAO or AO) with results in literature. We added results that we considered to be new:*

*Although the East Baltic Sea region is downstream from North Atlantic and Greenland – Baffin Bay region is upstream, after removing the general atmospheric circulation influence (we used NAO and AO indices) there still remained significant correlations between parameters of Baffin Bay region and the East Baltic Sea region, except winter.*
*We showed correlation coefficients between parameters of the Baffin Bay region and the East Baltic Sea region, which was the first time precisely for this region. We hope there will be revealed more physical mechanisms than we were able to reveal this time. This could help long period climate forecast to be more precise in the Eastern Baltic Sea region.*

c) Regarding the lagged correlations it would be nice to see how the authors connect their study to those discussing the persistence effect, such as Kolstad et al. (2015).

*Kolstad et al. (2015) investigated the persistence of European surface temperature and found that once the persistent weather patterns appear (e.g. temperature anomalies of at least one standard deviation above or below climatology in a month), then the persistence is observed irrespective of the data source or driving mechanisms, and the temperature itself is a more skilful predictor of the temperatures one month ahead. We concentrated on the analysis of the teleconnection between the Eastern Baltic Sea region and the Arctic region. The analysis of the temperature persistence of one region was beyond our scope.*

d) Also I found the maps in the figures are too small and they are very difficult to analyse. Some of them mostly repeat each other. For example q1000 and t1000 show very similar patterns. I wonder if it is possible to reduce the number of maps.

*We improved figures quality and left out wind speed at 1000 hPa (which is possible to assume from the geopotential heights) and specific humidity at 1000 hPa (which is very similar to temperature at 1000 hPa) to avoid redundancy.*

e) Technical comment: the paper by Lehmann et al., 2011 is referred to in the text but not listed in the literature section.

*Corrected.*

*The references you suggested are added to introduction, data and discussion paragraphs .*

Thank you once more for your trouble!

Sincerely yours,

    Liisi Jakobson
    Erko Jakobson
    Piia Post
    Jaak Jaagus

**References**

• Kolstad, E.W., Sobolowski, S.P., and Scaife, A.A.: Intraseasonal persistence of European surface temperatures, J CLIMATE, 28:5365–5374. doi:10.1175/JCLI-D-15-0053.1, 2015.

• Sato, K., Inoue, J., and Watanab, M.: Influence of the Gulf Stream on the Barents Sea ice retreat and Eurasian coldness during early winter, ENVIRON RES LETT, 9, 084009, 8pp, doi:10.1088/1748-9326/9/8/084009, 2014.

• Wu, B. Y., Zhang, R. H., D'Arrigo, R. et al.: On the Relationship between winter sea ice and summer atmospheric circulation over Eurasia, J CLIMATE, 26:5523-5536, doi:10.1175/JCLI-D-12- 00524.1, 2013.

---

## Author Comment (AC2)

**Answers to the referee 2                                23-Jun-2017**

Thank you very much for your competent and creative comments.  Please find below your comments repeated again and our answers. With the help of your advices, we have prepared a new version of our manuscript.

General Comments:

1. The thrust of the paper implies that the presence of statistical correlation implies causation, which is not the case. It is important for the authors to further explore the identified relationships by placing them in a climatological context and examining various potential atmospheric processes that may help explain the correlation results.

*To have a more focused paper we reduced the number of parameters, for that we made a general table of correlations with all our parameters and then chose only 3 for subsequent analysis: temperature, SLP and we added geopotential heights. We separated cold and warm winters (based on Baffin Bay region), similar to Sato et al, (2014); and added following analysis to reveal possible physical mechanisms why the Baltic Sea and the BB winters are in opposite phase relying on 1000 hPa temperature. We look atmospheric circulation differences using SLP, 700 hPa and 500 hPa geopotential height differences between warm and cold winters. We added also a cross-section of geopotential heights (up to 100 hPa) along the 60W vertical slice and plots of annual evolution of 500-hPa height differences at 60N, 70N and 75N (similar to Wu et al., 2013). See figures below:*

[Figure]

Figure 1. Seasonal difference maps (years with mild winters years with cold winters) in air temperature at 1000 hPa level (shading with confidence level of 95%), and (b) geopotential height at 500hPa level (contours).

[Figure]

Figure 2. Evolution of 500-hPa height differences between mild and cold winters at 60N; red and blue shading indicates differences at the 95% significance levels for positive and negative height, respectively.

[Figure]

Figure 3. Differences in the mean heights between mild and cold winters along the 60W vertical slice. Contour intervals are 10 gpm; blue represent negative height differences and red positive height differences.

*In discussion paragraph we added:*

*The large scale atmospheric circulation pattern in Figure 1 shows that the geopotential heights of 500 hPa are more than 100 gpm higher in mild winters than in cold ones, and the maximum of this height anomaly is centred over the maximum of the 1000 hPa temperature difference. It means that the whole column (up to 500 hPa) of the air in the Baffin Bay region is warmer than at cold years. Coming down to the lower surfaces (700 hPa, not shown), the maximum height anomaly is shifted to the east, what could be due to warmer sea surface of the Northern Atlantic compared to the regions that lay to west of it. The positive temperature anomaly (with the 500-hPa height anomalies) shifts towards east during the next seasons, reaching to Scandinavia/Baltic Sea region in summer (Figure 2). By Wu et al (2013) proposed mechanism, that associates the summer atmospheric circulation anomalies in the northern Eurasia with the previous winter ice conditions west of Greenland, supports our idea.*
*Figure 3 exhibit baroclinic structure of spring atmosphere north of 55N due to positive height anomalies in the lower troposphere below the 850 hPa and with further higher the negative ones. Similarly to Wu et al (2013) the vertical distribution of spring height anomalies differs from that of the previous winter when height anomalies show dominantly quasi-barotropic structure (not shown). With regression analysis they show the validity of their hypothesis of eastward propagation of the 500 hPa height anomalies. The same could be followed from Figure 2, where the evolution of 500 hPa height differences between mild and cold winters at 60 N is presented. Also at 65 N the similar pattern is present. At higher latitudes (70N and 75 N) this kind of signal propagation is missing.*

2. The authors present a great amount of results that need to be better interpreted, synthesized and placed in to a climatological/ atmospheric context supported by existing literature.

*To reduce the number of correlations we made a general table with all our parameters and then chose only 3 for subsequent analysis (temperature, SLP and we added height of geopotentials). We made extra analyses and supported our results with existing literature (see previous answer).*

3. The authors use simple linear correlation analyses to explore atmospheric teleconnections. I assume that they are speaking of the Pearson Correlation Coefficient. I have some concerns about this given that the areas of concern are in middle-to-high latitudes where teleconnections are known to be of non-linear nature. Also, the correlation method is applied to climate parameters such as wind and specific humidity that may not be normally distributed and significantly influence the results.

*We added the word "Pearson" to clarify which correlation we use in the manuscript. Teleconnections (like most physical processes) can often have non-linear nature, but until the process real relation functions are unknown, linear estimates are the most reasonable ones. We added to the text: "in this paper we use only linear correlations, non-linear correlations are not included".*
*To be statistically correct, our methods indeed assume normal distributions for all inputs. Still, as we are seeking not exact numbers but rather general patters, small violation of normal distribution assumptions should not have considerable effect. Also – as we use mostly seasonal*

*mean values – central limit theorem also gives us credit to assume that our data is at least in some extent normally distributed.*

4. The entire Baltic Sea region is represented by one single station located in southern Estonia (TP). The authors claim that the information provided in Figure 1 (i.e., Correlations between air temperatures at this location with locations across the greater Baltic Sea region during various season) shows that TP's climate represents the climate of the greater region very well. This may be the case for surface temperature, but I strongly doubt that same would hold true for the other variables such as wind characteristics. This can be seen in Figure 2 for JJA, for instance.

*We reduced the parameters of analysis. Temperature at 1000 hPa, SLP and geopotential heights at 700 hPa and 500 hPa are analysed. For SLP we prepared a similar figure as for temperature in manuscript (see below Figure 4).*
*To be more precise we renamed our title as the 'Atmospheric teleconnections between the Arctic and the Eastern Baltic Sea regions'.*

[Figure]

Figure 4. Correlation maps of SLP for the testing point in the Eastern Baltic Sea region.

5. For their analyses, the authors chose four atmospheric variables including air temperature, specific humidity, wind speed, and sea level pressure. Why did they choose these variables and not just sea level pressure, or the more typical 700 hPa geopotential heights for exploring atmospheric teleconnections?

*We have left out specific humidity and wind speed and have added 700 hPa and 500 hPa geopotential heights.*

6. What methods were used to remove the trends from the data?

*For detrending, firstly we calculated linear trend (k) and intercept (b) for each parameter every season in every grid point. Using these parameters linear detrending was done also for each parameter every season in every grid point:*

$Y_i = X_i - (k \cdot y \quad + b - X_a \quad )$.

*We added the formula with explanations in the manuscript.*

7. What methods were used to assess statistical significance?

*We used F-test for testing the significance of correlations. For comparison of averages (difference between warm and cold winters, was not included in the previous version), we used t-test assuming equal variances.*

$$F = \frac{(N-2) \cdot R^2}{1 - R^2}$$

8. The overall manuscript is clearly written baring some oddities in grammar and general use of the English language. I would recommend a more careful proof-reading of the revised manuscript. Some (not all) recommendations are included below.

Specific Comments:

Line 90: The authors mention several atmospheric teleconnections including the AO, NAO, PDO, SCA, EA, and EA/WR but do not explain what each of these are and on what basis they were included in the conversation. They also do not explain why most these were discounted up front and not addressed again even in the discussion section.

*To expose the role of different teleconnection indices we reorganized the analysis of teleconnection indices as follows (based on the suggestions of our referees):*

*we explained our choices of indices based on geographical position of the centres of action of the teleconnection patterns in data paragraph (see the segment 1 beneath);*

*we added to Results paragraph the table about the influence of teleconnection indices to correlations between the Baffin Bay region and the Eastern Baltic Sea region (see the table 1 below);*

*we added the analysis of PEU and found that the strength of influence is larger than all other teleconnection indices except much more larger impact of AO and NAO (see table 1);*

*we added to our discussion paragraph a new segment about the role of teleconnection indices, the possible reasons why other indices showed much less impact than AO and NAO indices, based on literature: Uotila et al, 2015; Lim, 2015; Comas-Bru and McDermott, 2014; Vihma et al., 2014; Moore et al., 2013.*

**Table 1.** *The partial correlations of teleconnection indices between 1000 hPa temperature at TP and the Baffin Bay-Greenland region (20-80W; 55 – 80). Smaller (than regular) values show higher impact of the index.*

| index | DJF | MAM | JJA | SON |
|---|---|---|---|---|
| *regular* | *-0.41* | *-0.23* | *0.15* | *-0.02* |
| AO | **-0.07** | **-0.10** | 0.19 | 0.08 |
| NAO | **-0.10** | **-0.11** | 0.23 | 0.04 |
| PDO | -0.45 | -0.26 | 0.06 | -0.11 |
| CAI | -0.41 | -0.21 | 0.15 | -0.01 |
| PEU | -0.42 | -0.18 | 0.19 | -0.02 |
| EA | -0.43 | -0.27 | 0.06 | 0 |
| EA/WR | -0.41 | -0.22 | 0.12 | -0.12 |
| SCA | -0.25 | -0.23 | 0.21 | -0.01 |

*Segment 1 of new version:*
*"The teleconnection indices we applied in our analyses were chosen according to the possible influence due to the geographical position of the centres of action of the teleconnection patterns over the North-Atlantic-Eurasian region. The following indices were chosen: 1) The North Atlantic Oscillation (NAO), which is the dominant mode of atmospheric variability in the North Atlantic sector throughout the year (Barnston and Livezey, 1987); 2) The Arctic Oscillation (AO), which is usually defined as the first EOF of the mean sea level pressure field in the Northern Hemisphere (Ambaum et al., 2001); 3) The Scandinavian Pattern (SCA), which consists of a primary circulation centre over Scandinavia, with two other weaker centres of action with the opposite sign, one over the north eastern Atlantic and the other over central Siberia to the southwest of Lake Baikal (Bueh and Nakamura, 2007); 4) The East Atlantic Pattern (EA), which consists of a north-south dipole of anomaly centres spanning the North Atlantic from east to west (Barnston and Livezey, 1987); 5) The East Atlantic/West Russia Pattern (EA/WR), which consists of four main anomaly centres: Europe, northern China, central North Atlantic and north of the Caspian Sea; 6) The Polar/ Eurasia Pattern (PEU) consists of height anomalies over the polar region, and opposite anomalies over northern China and Mongolia.; 7) Additionally, Pacific Decadel Oscillation (PDO), which is the dominant year-round pattern of monthly North Pacific sea surface temperature (SST) variability was included. Although its geographical centres are far from the Baltic Sea region, Uotila et al (2015) found that PDO correlated significantly with the ice concentration and temperature of Baltic Sea. All indices were downloaded from the NOAA-CPC database (http://www.cpc.noaa.gov)."*

Line 105: The authors mention that they detrended the seasonal time series "to avoid the correlations to be caused by mutual trends in input variables." They also claim that the detrended and original correlation results were very similar. For this reason, they only show correlation results from "regular data". The results surprise me (i.e., similar correlations from original and detrended data), especially given the large recent temporal trends in many of the variables that are explored (i.e.,temperature)in the high latitudes of the northern hemisphere. It is also important to note that the conclusions regarding teleconnections that one can reach from the original series versus detrended series may be different. Are the authors exploring the connections that include long term climatic trends such as global warming, or are they interested in understanding the relationships as they may exist independently of such trends?

*Thank you especially for the last sentence, it ended our hesitations should we present results with or without trend. In the upgraded version, we show only results without detrending, to focus connections that are present in our world that is influenced by global climate change*

*trends. We include discussion about detrended data to clarify that presented correlations are not because of trends. Our sentence "differences between the areal averages of correlations were up to 0.02 in both directions" is indeed a bit misleading, as there are small regions where the difference is larger than 0.4, we replaced it with "detrending did not change general patterns of correlations with TP, only intensified negative correlation in the Greenland region ".*

Line191: The authors claim that"...,the winter mean temperature is not dependent on weather conditions during the previous seasons." But on line 199 they proceed to make the following claim: "Winter temperature at the TP has a strong negative correlation in the Taimyr region in the previous summer." To me, these statements seem to contradict themselves.

*Thank you for asking, there was indeed conflict between these sentences. We upgraded the text as follows: " At the same time, the winter mean temperature has almost no dependent on weather conditions during the previous seasons, there is only small region with strong negative correlation in the Taimyr region in the previous summer (lag=6).*

Line 235: The authors state that "To avoid false correlations, only the results that were present in both the regular and the detrended data were discussed." I am not sure what is meant by "false correlations". Like I mentioned earlier, detrended data for instance, may hold a different story, not a false story.

*You are correct, we just remove this sentence (we explained reasons two comments above).*

Line 25: find another word for "disconfirm"

*We replaced "disconfirm" with "disagree".*

Line 26-27: It is not clear what "both" is referring to in the sentence starting with "They found that from...."

*We changed the sentence as follows: "They found that from October to December, the main factors responsible for the Arctic deep tropospheric warming are: 1) the recent decadal fluctuations and 2) long-term changes in sea surface temperatures. These two factors are located outside the Arctic."*

Line 33: "Arctic amplification" should be Arctic Amplification

*Corrected*

Line 67: It is not customary for sentences to begin with "But"

*We changed the sentence as follows:*

*There is no clear understanding about the reasons for the changes in these indices or climatic parameters in the Baltic Sea region in most recent time.*

Line 68: I would suggest replacing "last decades" with most recent?

*Corrected.*

Line 71: Rework the sentences starting with "Therefore, our aim is to....

*We changed the segment as follows: "Our aim is to clarify how the climatic parameters in the Eastern Baltic Sea and Arctic regions are associated. Knowledge of such connections helps to define regions in the Arctic that could be with higher extent associated with the Baltic region climate change."*

Line 123: Replace the word "huge" with large

*Corrected*

Line 132: Can the word "distinguished" be replaced with different or distinct?

*Replaced with distinct.*

Thank you once more,

Sincerely yours,

Liisi Jakobson
Erko Jakobson
Piia Post
Jaak Jaagus

**References** (If we use in our answers references that were already given in our article then we will not give the reference here again):

- Barnston, A. G., and Livezey, R.E.: Classification, seasonality and persistence of low-frequency atmospheric circulation patterns, MON WEATHER REV, 115, 1083-1126, 1987.
- Bueh, C. and Nakamura, H.: Scandinavian pattern and its climatic impact, Q J ROY METEOR SOC. 133: 2117 – 2131, DOI: 10.1002/qj.173, 2007.
- Comas-Bru, L. and McDermott, F.: Impacts of the EA and SCA patterns on the European twentieth century NAO–winter climate relationship. Q J ROY METEOR SOC, 140: 354–363. doi:10.1002/qj.2158, 2014.
- Lim, Y.K.: The East Atlantic/West Russia (EA/WR) teleconnection in the North Atlantic: climate impact and relation to Rossby wave propagation, CLIM DYNAM, 44: 3211. doi:10.1007/s00382-014-2381-4, 2015.
- Moore, G.W.K., Renfrew, I.A., Pickart, R.: Multi-decadal mobility of the North Atlantic Oscillation. J CLIMATE. 26 : 2453–2466, DOI:10.1175/JCLI-D-12-00023.1, 2013.
- Sato, K., Inoue, J., and Watanab, M.: Influence of the Gulf Stream on the Barents Sea ice retreat and Eurasian coldness during early winter, ENVIRON RES LETT, 9, 084009, 8pp, doi:10.1088/1748-9326/9/8/084009, 2014.
- Uotila, P., Vihma, T., and Haapala, J.: Atmospheric and oceanic conditions and the extremely mild Baltic Sea ice winter 2014/15, GEOPHYS RES LETT, doi:10.1002/2015GL064901, 2015.
- Vihma, T., Cheng, B., and Uotila, P.: Linkages between Arctic sea ice cover, large-scale atmospheric circulation, and weather and ice conditions in the Gulf of Bothnia, Baltic Sea, Advances in Polar Science, 25(4), 289-299, doi: 10.13679/j.advps.2014.4.00289, 2014.
- Wu, B. Y., Zhang, R. H., D'Arrigo, R. et al.: On the Relationship between winter sea ice and summer atmospheric circulation over Eurasia, J CLIMATE, 26:5523-5536, doi:10.1175/JCLI-D-12- 00524.1, 2013.

---

## Author Response (AR2)

Thank you very much for your competent comments. Please find below your comments repeated again and our answers. With the help of your advices, we have prepared a new version of our manuscript.

1. The description of the background of Arctic-midlatitude linkages and possible physical relationships between Arctic climate change and midlatitude weather and climate has been improved, but still needs some revision to avoid sketchy physical explanations and inconsistencies. To illustrate this, I give two examples here:

(a) The paragraph L59-L75 is difficult to follow. I would ask for a better and more systematic view on the studies on Arctic-midlat linkages considering that the suggested processes depend on season, geographical region and other impacts.

*We reorganized the segment as follows:*

*The linkages between the Arctic and midlatitudes depend on geographical region, season and other impacts. There are certain geographical regions in the Arctic that have greater amount of warming and the influence of these is more investigated. Arctic warming over the Barents and Kara Seas and its impacts on the mid-latitude circulations have been widely discussed (Dobricic et al., 2016; Semenov and Latif, 2015; Kug et al., 2015; Sato et al., 2014). Another particular regional warm core (Screen and Simmonds, 2010) is the East Siberian and Chukchi Seas, which is related to severe winters over North America (Kug et al., 2015; Lee et al., 2015). Screen and Simmonds (2010) brought out also the third particular regional warm core – northeast Canada and Greenland which has been less investigated. Wu et al., (2013) focused on winter sea ice concentration west of Greenland, including the Labrador Sea, Davis Strait, Baffin Bay, and Hudson Bay and found that winter sea ice concentration west of Greenland is a possible precursor for summer atmospheric circulation and rainfall anomalies over northern Eurasia. If we look at the regions in the mid-latitudes then potential Arctic teleconnections with Europe are less clear than with North America and Asia (Overland et al., 2015). The linkages between the Arctic and midlatitudes depend also on season. Summer is exceptional season when the weather conditions are less affected by large-scale atmospheric circulation both in midlatitudes and in the Arctic. But the influence of the increase in late summer open water area is directly contributing to a modification of large scale atmospheric circulation patterns (Overland and Wang, 2010).*

(b) The authors claimed "Common supposition is that sea ice declines are primarily responsible for amplified Arctic tropospheric warming" (L62) In my understanding, that is not the case. There are various feedback processes, not only the ice-albedo feedback, which contribute to amplified warming in the Arctic. These are feedback effects associated with temperature, water

vapour and clouds. Some studies suggest that temperature feedbacks are the main contributors to Arctic amplification.

*We reorganized the whole segment and left out the supposition on L62.*

2. I can accept that the authors did not apply the more demanding criterion of reproducibility. But still, the authors tested the Nullhypothesis of no correlation only. I think, this needs to be expanded by, at least, taking into account the reduction of degrees of freedom due to autocorrelation and also by estimating the confidence intervalls of the correlation coefficients. Furthermore, I suggest to include the discussion of detrended correlation analysis.

*Thank you very much for the idea of reduction of degrees of freedom. Indeed, this was something we had not taken into account. We upgraded Figures 2 and 6 using reduced degrees of freedom. Confidence intervals were not added. For figures, we could add 4 extra figures with statistically minimal and maximal values but still – the best estimate of correlations that are statistically significant means, that shown positive correlations are positive and negative correlations are negative, exact interval values are not important for our results. For tables of average correlations the intervals would be also confusing, as there are regional variations in both in correlation and in correlation variations, so every kind of averaging would be imprecise.*

*We had discussion of detrended correlation analysis in the first version, but as reviewers suggested, we continued with only result with correlations with trends. We are not interested in teleconnections, that there could be if there would be no trends in the climate.*

3. To follow the discussion and the conclusion, maps of the partial correlation coefficients (for AO/NAO and SCA influence) have to be shown.

*To compress the manuscript we have been removing (according comments from the first round) these figures and showed only small Table 2 with the most important information. Now we added only some references to the Table 2 in the discussion and conclusion paragraph. We hope it will help to follow the manuscript.*

4. I appreciate the new section 3.3, but the shown figures need more careful interpretation. E.g., the authors claimed at L242-244 " The annual evolution of 500 hPa height differences at 60°N shows that the positive temperature anomaly at the Greenland sector shifts towards east during the next seasons, reaching to Scandinavia/Baltic Sea region in summer (Figure 5)." I only see an eastward shift until spring, afterwards there is an interruptions, indicated by the negative differences around May at 0-40degrees E.

*We added a following sentence:*

*The propagation of the mid-tropospheric anomalies in this region is nonlinear: these height anomalies are significant only over some areas and months and in May they are slightly negative.*

5. Another example for sketchy physical statements: L248 "There is a large inertia in the atmosphere causing lag effects." Typical atmospheric time-scale is about 10-14 days. For lag correlation with 3 or more months, other processes may play a role. Such processes have to be discussed.

*We improved the beginning of this paragraph as follows:*

* The climate system consists of various interactive components that have highly various response times. The estimated time scales in atmosphere grow with height and reach up to months, but due to atmospheric interactions with the oceans and cryosphere, the  conditions in atmosphere may have even longer response times. *

6. L308: The mechanism, proposed by Wu et al., 2013, has to be explained.

*We added the following explanation:*

*According to Wu et al. (2013) the summer atmospheric circulation anomalies in the northern Eurasia are associated with the previous winter SIC west of Greenland. The mechanism is based on horseshoe-like pattern of SST anomalies in the North Atlantic that persist in winter and spring. Such anomaly impacts on ensuing spring atmosphere over the North Atlantic which links winter-spring SIC and SST anomalies and summer atmospheric circulation anomalies over northern Eurasia including the Baltic Sea region. This proposed mechanism supports our results.*

7. Conclusions: 5 findings are given, but I do not find evidence for second, third and forth findings (L322-330) in the manuscript.

*We believe that these findings should be covered as follows:*

*Second – indeed, the figures are not shown (at first we had them but then removed according to suggestions of referees), but the results are given in paragraph 3.2;*

*Third – please, look at the table 1;*

*Fourth – we consider Figure 5 to be the evidence of forth finding.*

**New Reference:**

Overland, J., and Wang, M.: Large-scale atmospheric circulation changes associated with the recent loss of Arctic sea ice. TELLUS, 62A:1-9, doi: 10.1111/j.1600-0870.2009.00421.x, 2010.

Thank you once more,

Sincerely yours,

Liisi Jakobson

Erko Jakobson

Piia Post

Jaak Jaagus